# Diss-l-ECT: Dissecting Graph Data with Local Euler Characteristic Transforms

**Julius von Rohrscheidt** [1 2]   **Bastian Rieck** [1 2 3]

## Abstract

The Euler Characteristic Transform (ECT) is an efficiently-computable geometrical-topological invariant that characterizes the *global* shape of data. In this paper, we introduce the *Local Euler Characteristic Transform* ($\ell$-ECT), a novel extension of the ECT particularly designed to enhance expressivity and interpretability in graph representation learning. Unlike traditional graph neural networks (GNNs), which may lose critical local details through aggregation, the $\ell$-ECT provides a lossless representation of local neighborhoods. This approach addresses key limitations in GNNs by preserving local structures while maintaining global interpretability. Moreover, we construct a rotation-invariant metric based on $\ell$-ECTs for spatial alignment of data spaces. Our method exhibits superior performance compared to standard GNNs on a variety of benchmark node-classification tasks, while also offering theoretical guarantees that demonstrate its effectiveness.

## 1. Introduction

Traditionally, graph neural network (GNNs) rely on message-passing schemes to aggregate node features. While effective for many tasks, this approach often leads to the loss of critical *local* information, as the aggregation process can diffuse and obscure the original node vector representations (Topping et al., 2022). This limitation makes it challenging to preserve local characteristics that may be essential for some applications. To address this limitation, we harness the *Euler Characteristic Transform* (Turner et al., 2014, ECT), an expressive geometrical-topological invariant that can be computed efficiently. Relying only on (weighted) sums, the ECT can be computed efficiently, making it a powerful tool for representation learning (Röell & Rieck, 2024). Moreover, the ECT is known to be *invertible* for data in $\mathbb{R}^n$, ensuring that the original data can always be reconstructed (Curry et al., 2022; Ghrist et al., 2018). In this paper, we extend the ECT to *local* neighborhoods, presenting the *Local Euler Characteristic Transform* ($\ell$-ECT), a novel method designed to preserve *local* structure while retaining *global* interpretability. The $\ell$-ECT captures both topological (i.e., structural) and geometrical (i.e., spatial) information around each data point, making it particularly advantageous for graph-based or higher-order data.[1] The $\ell$-ECT thus becomes an expressive *fingerprint* of local neighborhoods, specifically addressing the challenge of neighborhood aggregation in featured graphs while ensuring the *lossless representation* of local node neighborhoods. We theoretically investigate how $\ell$-ECTs maintain critical local details, and therefore provide a nuanced representation that can be used for downstream graph-learning tasks such as node classification. Our method is highly effective, particularly for tasks where node feature aggregation may obscure essential differences, such as in graphs with high heterophily. Additionally, the $\ell$-ECT framework's natural vector representation makes it compatible with a wide range of machine-learning models, facilitating both performance and interpretability.

**As our main contributions,** we (i) construct $\ell$-ECTs in the context of embedded simplicial complexes (and graphs) and theoretically investigate their expressivity in the special case of featured graphs, (ii) empirically show that this expressivity positions $\ell$-ECTs as a powerful general tool for interpretable node classification, often superior to standard GNNs, and (iii) introduce an efficiently computable rotation-invariant metric based on $\ell$-ECTs that facilitates the spatial alignment of geometric graphs.

## 2. Background

We define our method in the most general setting, i.e., that of a simplicial complex, while also providing a brief introduction to graph neural networks.

---

[1]Institute of AI for Health, Helmholtz Munich, Germany [2]Technical University of Munich, Germany [3]University of Fribourg, Switzerland. Correspondence to: Julius von Rohrscheidt <julius.rohrscheidt@helmholtz-munich.de>, Bastian Rieck <bastian.grossenbacher@unifr.ch>.

*Proceedings of the $42^{nd}$ International Conference on Machine Learning*, Vancouver, Canada. PMLR 267, 2025. Copyright 2025 by the author(s).

[1]Our experiments deliberately focus on graphs, but we note that the method can be extended to novel higher-order datasets based on *simplicial complexes*, for instance (Ballester et al., 2025).

**Simplicial Complexes** A *simplicial complex* $K$ is a mathematical structure that generalizes graphs to model higher-order (non-dyadic) relationships and interactions. While graphs model pairwise (dyadic) connections between entities using nodes and edges, simplicial complexes extend this representation to higher dimensions by including triangles (2-simplices), tetrahedra (3-simplices), and their higher-dimensional analogues. Let $v_0, \ldots, v_k \in \mathbb{R}^n$ be *affinely independent* points. The *(geometric) k-simplex* determined by these vertices is the convex hull

$$\sigma = [v_0 v_1 \cdots v_k] := \left\{ \sum_{i=0}^{k} \lambda_i v_i \,\middle|\, \lambda_i \geq 0, \, \sum_{i=0}^{k} \lambda_i = 1 \right\}.$$

The points $v_0, \ldots, v_k$ are called the *vertices* of $\sigma$. Simplices determined by a subset of $v_0, \ldots, v_k$ are called *faces* of $\sigma$. Formally, a simplicial complex is a finite collection of simplices such that every face of a simplex in the collection is also in the collection, and the intersection of any two simplices is either empty or a common face.

**Euler Characteristic** The *Euler characteristic* $\chi$ is a topological invariant that provides a summary of the "shape" or structure of a topological space, such as a simplicial complex. It is defined as the alternating sum of the number of simplices in each dimension, i.e.,

$$\chi(K) = \sum_{k=0}^{d} (-1)^k \sigma_k(K), \tag{1}$$

where $\sigma_k(K)$ is the number of $k$-dimensional simplices in the simplicial complex $K$, and $d$ is the dimension of $K$. As a topological invariant, the Euler characteristic remains unchanged under transformations like homeomorphisms, making it—despite its simplicity—a fundamental tool for distinguishing topological spaces.

**Graph Neural Networks and Message Passing** *Graph neural networks* (GNNs) are a class of neural network models designed to operate on graph-structured data. They extend neural networks by incorporating the relational structure inherent to graphs, enabling the learning of tasks such as *node classification*. The core mechanism of many GNNs is *message passing*, an iterative procedure that propagates information through the graph to update node representations based on their local neighborhood. Given a graph $G = (V, E)$, where $V$ is the set of nodes and $E$ is the set of edges, each node $v \in V$ is associated with a *feature vector* $\mathbf{x}_v$. At each layer $t$, i.e., each message-passing step, a new node embedding $\mathbf{h}_v^{(t+1)}$ is calculated via

$$\text{UPDATE}\left(\mathbf{h}_v^{(t)}, \text{AGG}\left(\{\mathbf{h}_u^{(t)} \mid u \in \mathcal{N}(v)\}\right)\right), \tag{2}$$

where $\mathcal{N}(v)$ denotes the set of neighbors of node $v$, and AGG and UPDATE are learnable functions parameterized by the model. The AGG function combines information from neighboring nodes, while the UPDATE function refines node embeddings. Popular choices for these functions include mean, sum, and attention mechanisms. Through multiple layers of message passing, GNNs aggregate information from larger neighborhoods, capturing both local and global graph structure (Veličković, 2023).

## 3. Related Work

GNNs have revolutionized the field of graph representation learning by enabling end-to-end learning of node/graph embeddings through message passing (Kipf & Welling, 2017). However, traditional GNNs face theoretical limitations that pose fundamental obstructions to learning expressive and general representations of graph data (Xu et al., 2019). Related to the latter phenomenon, GNNs are known to suffer from issues like *oversmoothing* (Rusch et al., 2023; Zhang et al., 2024) and *oversquashing* (Di Giovanni et al., 2023). Hamilton et al. (2017) and Veličković et al. (2018) have addressed these issues by incorporating sampling and attention mechanisms into the message-passing paradigm. However, even these advancements often show limited performance, particularly in graphs with high heterophily, and there is no "general" GNN capable of handling both heterophilous and homophilous graphs. Recent work in graph machine learning thus started incorporating additional inductive biases into architectures, such as geometric information (Joshi et al., 2023; Pei et al., 2020; Southern et al., 2023) or topological information (Horn et al., 2022; Verma et al., 2024), with the ultimate goal of improving the *expressivity* of a model, i.e., its capability to distinguish between non-isomorphic families of graphs (Morris et al., 2023).

Many such endeavors arise from the field of *topological deep learning* (Papamarkou et al., 2024), which aims to develop models that are "aware" of the underlying topology of a space, and thus also capable of handling data with higher-order relations. Other constructions include special architectures for heterophilous tasks that are *not* based on message-passing (Lim et al., 2021), or *modifications* of the graph itself to improve predictive performance. Suresh et al. (2021), for instance, use edge rewiring to raise graph assortativity and thus gain accuracy under low homophily, whereas Luan et al. (2022) mix feature channels during aggregation to obtain state-of-the-art results on heterophilous benchmarks. Finally, as Rampášek et al. (2022) show, a hybrid model, combining message-passing (local information) with attention (global information) via structural encodings, may exhibit high expressivity and high scalability. Subsequently, Müller et al. (2024) extended these results by providing a taxonomy of elements related to the "design space" of graph transformers.

As a geometrical-topological invariant, the ECT is poised to contribute to more expressive architectures. Contributing Being already a popular tool in topological data analysis (Ghrist et al., 2018; Turner et al., 2014), recent extensions started tackling the integration into deep-learning architectures (Röell & Rieck, 2024) or the incorporation of additional invariance properties (Curry et al., 2022; Marsh et al., 2024). Despite advantageous performance in shape-classification tasks, however, *all* existing contributions solely focus on *global* ECTs and do not discuss any *local* aspects, which are crucial for our approach. In addition to the ECT, some works also use other topology-based tools in graph learning, primarily *persistent homology* (PH), an expressive but computationally expensive geometrical-topological invariant. Examples of this approach include Rieck et al. (2019) and Hofer et al. (2020), who use PH for graph classification, or Zhao & Wang (2019), who learn a weighted kernel on topological descriptors arising during PH computations. Closest to our approach in spirit is Zhao et al. (2020), who include topological features of graph neighborhoods into a GNN, again leveraging PH. However, to the best of our knowledge, ours is the first work to develop *local* variants of the ECT for graph-learning tasks, analyze the theoretical properties of such local variants, and finally show their empirical utility for node classification.

## 4. Methods

**Euler Characteristic Transform (ECT)** The *Euler Characteristic Transform* (ECT) of a simplicial complex $X \subset \mathbb{R}^n$ is a function $\mathrm{ECT}(X) \colon S^{n-1} \times \mathbb{R} \to \mathbb{Z}$, given by

$$\mathrm{ECT}(X)(v, t) := \chi(\{x \in X \mid x \cdot v \leq t\}), \qquad (3)$$

where $\chi$ denotes the Euler characteristic and $x \cdot v$ denotes the Euclidean dot product. The interpretation of $\mathrm{ECT}(X)$ is that it scans the ambient space of $X$ in every direction and records the Euler characteristic of the sublevel sets. The $\mathrm{ECT}(X)$ is *invertible*, meaning that $X$ can be recovered from $\mathrm{ECT}(X)$, as long as $X$ is a so-called constructible set (Curry et al., 2022; Ghrist et al., 2018). The main focus of this work are *compact geometric simplicial complexes* (like geometric graphs), which are constructible, and thus the invertibility theorem applies in our setting. Note that in practice, we *approximate* $\mathrm{ECT}(X)$ via $\overline{\mathrm{ECT}}(X)_{(m,l)} := \mathrm{ECT}(X)_{|\{v_1,\ldots,v_m\} \times \{t_1,\ldots,t_l\}}$ for uniformly-distributed directions $v_1, \ldots, v_m \in S^{n-1}$ and filtration steps $t_1, \ldots, t_l \in \mathbb{R}$. Since $X$ is compact, $t_1, \ldots, t_l$ can be chosen to lie in a compact interval $[a, b]$ with $t_1 = a$ and $t_l = b$, and so that the sequence $\{t_i\}_i$ forms a uniform partition of $[a, b]$. We note that this approximation is efficiently computable and has a natural representation as a vector of dimension $m \cdot l$. Regarding the choice of the magnitudes of $m, l$ we notice that the *expected* nearest-neighbor distance for uniform samples on $S^n$ scales as

$\mathcal{O}((\log m / m)^{1/n})$ (Beck, 1987), and that the equidistant partitioning of a compact interval scales as $\mathcal{O}(1/l)$, leading to $\mathcal{O}((\log m / m)^{1/(n-1)} l^{-1})$ for the total approximation error of the domain of $\mathrm{ECT}(X)$. Curry et al. (2022) prove that the aforementioned approximation actually determines the *true* value, provided that $m, l$ are sufficiently large. We notice that both translations and scalings of $X$ in the ambient space lead to a reparametrization of $\mathrm{ECT}(X)$. Hence, $\mathrm{ECT}(X)$ remains essentially unaltered (up to a parameter change) under these two types of transformations.

**Local ECT ($\ell$-ECT)** Given a geometric simplicial complex $X \subset \mathbb{R}^n$ and a vertex $x \in X$, we define the *local ECT* of x with respect to $k \geq 0$ as

$$\ell\text{-}\mathrm{ECT}_k(x; X) := \mathrm{ECT}(N_k(x; X)), \qquad (4)$$

where $N_k(x; X)$ denotes an appropriate *local neighborhood* of $x$ in $X$, whose locality scale is controlled by a parameter $k$. Usually, $N_k(x; X)$ will be either the full subcomplex of $X$, which is spanned by the $k$-hop neighbors of $x$, or the full subcomplex of $X$, which is spanned by the $k$-nearest vertices of $x$. The first important special case arises when $X$ is a 0-dimensional simplicial complex, i.e., a point cloud. In this case, the full subcomplex of $X$, which is spanned by the $k$-nearest vertices of $x$, $N_k(x; X)$, is given by the $k$-nearest neighbors of $x$. Being based on the Euler Characteristic, the construction of $\ell$-ECTs appears to be purely topological at first glance. However, in light of the invertibility theorem, we note that $\ell\text{-}\mathrm{ECT}(x; X)$ can be interpreted as a *fingerprint* of a local neighborhood of $x$ in $X$. The upshot is that this fingerprint can be well approximated in practice, making it possible to obtain *local representations* of combinatorial data embedded in Euclidean space. Similar to the approximation of the ECT, this approximation works by sampling $v_1, \ldots, v_m \in S^{n-1}$ and $t_1, \ldots, t_l \in \mathbb{R}$, and considering $\overline{\mathrm{ECT}}(N_k(x; X))_{(m,l)}$, instead of $\ell\text{-}\mathrm{ECT}_k(x; X)$. The latter quantity is well-computable in practice, and the approximation error can be controlled by the sample sizes $m$ and $l$, as we discussed above. Again, this approximation has a natural representation as a vector of dimension $m \cdot l$, enabling us to encode local structural information of point neighborhoods in an approximate lossless way that can readily be used by machine-learning algorithms for downstream tasks.

### 4.1. Properties of $\ell$-ECTs

Our formulation of $\ell$-ECTs provides a natural representation of local neighborhoods of geometric simplicial complexes. One important special case is that of *featured graphs*, meaning graphs in which every node admits a feature vector. The latter data structure forms the basis of many modern graph-learning tasks, such as node classification, graph classification, or graph regression.

The predominant class of methods to deal with these graph learning problems are message-passing graph neural networks. We develop an alternative procedure for dealing with featured graph data, built on $\ell$-ECTs and we show that $\ell$-ECTs provide sufficient information to perform message passing, which we explain in the following.

**Definition 1.** *A* featured graph *is a (non-directed) graph* $\mathcal{G}$ *such that every node* $v \in \mathcal{G}$ *admits a feature vector* $x(v) \in \mathbb{R}^n$. *We denote the set of nodes of* $\mathcal{G}$ *by* $V(\mathcal{G})$, *and the set of edges by* $E(\mathcal{G})$.

We notice that a featured graph $\mathcal{G}$ can naturally be interpreted as a graph embedded in $\mathbb{R}^n$, by representing each node feature vector as a point in $\mathbb{R}^n$, and by drawing an edge between two embedded points if and only if there is an edge between the underlying nodes in $\mathcal{G}$. This construction yields a graph isomorphism between $\mathcal{G}$ and the embedded graph if and only if for any pair of nodes $v, w \in \mathcal{G}$ with $v \neq w$ we have $x(v) \neq x(w)$ for their associated feature vectors. In practice, the latter assumption can always be achieved by adding an arbitrarily small portion of Gaussian noise to each feature vector, and we therefore may restrict ourselves to featured graphs that yield an isomorphism on their Euclidean embeddings.[2] We now show that $\ell$-ECTs are in fact expressive graph-learning representations.

**Theorem 1.** *Let* $\mathcal{G}$ *be a featured graph and let* $\{\ell\text{-ECT}_1(x; \mathcal{G})\}_x$ *be the collection of local* ECT*s with respect to the 1-hop neighborhoods in* $\mathcal{G}$. *Then the collection* $\{\ell\text{-ECT}_1(x; \mathcal{G})\}_x$ *provides the necessary (non-learnable) information for performing a single message-passing step on* $\mathcal{G}$, *in the sense that for a given vertex* $x \in \mathcal{G}$ *one can reconstruct the feature vectors of its 1-hop neighborhood from* $\ell\text{-ECT}_1(x; \mathcal{G})$.

Theorem 1 tells us that for a featured graph $\mathcal{G}$, the collection $\{\ell\text{-ECT}_1(x; \mathcal{G})\}_x$ already contains sufficient information to perform a single step of message passing. The advantage of using $\ell$-ECTs instead of message passing to represent featured graph data lies in the possibility to *additionally* use $\{\ell\text{-ECT}_k(x; \mathcal{G})\}_x$ for $k \geq 2$, which contain both structural and feature vector information of larger neighborhoods of nodes in the graph. This type of information is typically *not* explicitly available through message passing since passing messages to non-direct neighbors depends on prior message passing steps, which solely produce an aggregation of neighboring feature vectors.

---

[2]Alternatively, we can drop the requirement of an embedding by noting that a featured graph can be considered as an abstract simplicial complex $\mathcal{G}$ with an *arbitrary* function $f \colon \mathcal{G} \to \mathbb{R}^n$ defined on its vertices and edges. In this case, we may define the ECT as $\text{ECT}(X)(v, t) := \chi(\{f^{-1}\{x \in \mathbb{R}^n \mid x \cdot v \leq t\})$. This formulation, developed by Marsh & Beers (2023), demonstrates that the ECT is *generally* applicable and does not require node features to provide an embedding of a graph.

In addition to essentially subsuming the information from one message-passing step, we can also show that the $\ell$-ECT is "aware" of local structures like subgraphs. As shown by Chen et al. (2020), message-passing graph neural networks *cannot* perform counting of induced subgraphs for *any* connected substructure consisting of 3 or more nodes. By contrast, we will now show that ECTs for featured graphs and their local variants can indeed be used to perform subgraph counting. We start with the definitions of the necessary concepts.

**Definition 2.** *Two featured graphs* $\mathcal{G}_1$ *and* $\mathcal{G}_2$ *are* isomorphic *if there is a bijection* $\pi \colon V(\mathcal{G}_1) \to V(\mathcal{G}_2)$, *such that* $(v, w) \in E(\mathcal{G}_1)$ *if and only if* $(\pi(v), \pi(w)) \in E(\mathcal{G}_2)$ *and so that for all* $v \in \mathcal{G}_1$ *one has* $x(v) = x(\pi(v))$ *for the respective feature vectors.*

A featured graph $\mathcal{G}_S$ is called a *subgraph* of $\mathcal{G}$ if $V(\mathcal{G}_S) \subset V(\mathcal{G})$ and $E(\mathcal{G}_S) \subset E(\mathcal{G})$, such that the respective node features remain unaltered under the induced embedding. A featured graph $\mathcal{G}_S$ is called an *induced subgraph* of $\mathcal{G}$, if $\mathcal{G}_S$ is a subgraph of $\mathcal{G}$, and if $E(\mathcal{G}_S) = E(\mathcal{G}) \cap \mathcal{G}_S$. For two featured graphs $\mathcal{G}$ and $\mathcal{G}_S$, we define $C_{\text{Sub}}(\mathcal{G}; \mathcal{G}_S)$ to be the number of subgraphs in $\mathcal{G}$ that are isomorphic to $\mathcal{G}_S$. Similarly, we define $C_{\text{Ind}}(\mathcal{G}; \mathcal{G}_S)$ to be the number of induced subgraphs in $\mathcal{G}$ which are isomorphic to $\mathcal{G}_S$.

**Theorem 2.** *Two featured graphs* $\mathcal{G}_1$ *and* $\mathcal{G}_2$ *are isomorphic if and only if* $\text{ECT}(\mathcal{G}_1) = \text{ECT}(\mathcal{G}_2)$.

An immediate consequence of the previous Theorem is:

**Corollary 1.** ECT*s can perform subgraph counting.*

We therefore conclude that ECT-based methods for graph-representation learning can be more powerful than message-passing-based approaches, suggesting the development of *hybrid* architectures, making use of both message passing *and* ECT variants.

### 4.2. Rotation-Invariant Metric based on Local ECTs

The aforementioned invariance properties of ECTs with respect to translations and scalings naturally raise the question if $\ell$-ECTs may be used to compare the local neighborhoods of two distinct points/vertices. However, the ECT is sensitive to rotations since rotating the underlying simplicial complex leads to a misalignment of the respective directions in $S^{n-1}$. Because a local comparison should *not* depend on the choice of a coordinate system, this property is a fundamental obstruction of using $\ell$-ECT as a local similarity measure. We therefore construct a novel *rotation-invariant metric* as follows. Let $X, Y \subset \mathbb{R}^n$ be two finite geometric simplicial complexes. Since $X, Y$ are finite, $\text{ECT}(X)$ and $\text{ECT}(Y)$ only take finitely many values, and we may define a similarity measure $d_{\text{ECT}}(X, Y)$ as

$$d_{\text{ECT}}(X, Y) := \inf_{\rho \in \text{SO}(n)} \|(\text{ECT}(X) - \text{ECT}(\rho Y))\|_\infty. \quad (5)$$

We first prove that this similarity measure satisfies the definitions of a metric.

**Theorem 3.** $d_{\mathrm{ECT}}$ *is a metric on the collection of rotation classes of finite simplicial complexes embedded in* $\mathbb{R}^n$.

Theorem 3 ensures that we may use $d_{\mathrm{ECT}}$ as a metric that measures the similarity between embedded simplicial complexes up to rotation. In particular, for a simplicial complex $X \subset \mathbb{R}^n$ and $x, y \in X$, we have a rotation-invariant measure to compare local neighborhoods of $x$ and $y$ by defining $d_{\mathrm{ECT}}^k(x, y; X)$ as $\inf_{\rho \in \mathrm{SO}(n)} \|\ell\text{-}\mathrm{ECT}_k(x; X) - \ell\text{-}\mathrm{ECT}_k(y; \rho X)\|_\infty$. In practice, we approximate $d_{\mathrm{ECT}}(X, Y)$ by

$$\inf_{\rho \in \mathrm{SO}(n)} \left\| \overline{\mathrm{ECT}}(X)_{(m,l)} - \overline{\mathrm{ECT}}(\rho Y)_{(m,l)} \right\|_\infty \quad (6)$$

for a choice of samples $v_1, \ldots, v_m \in S^{n-1}$ and $t_1, \ldots, t_l \in \mathbb{R}$; this works analogously for the local version $d_{\mathrm{ECT}}^k(x, y; X)$. As discussed before, the approximations of the ECTs used in Eq. (6) have a natural vector representation, so that the $\|\bullet\|_\infty$ in Eq. (6) is in fact the maximum of the entry-wise absolute differences between the two respective representation vectors. Hence, the approximation shown in Eq. (6) is *efficiently computable*. However, our experiments in Section 5 use the Euclidean metric for differentiability reasons.

### 4.3. Limitations

While $\ell$-ECTs present clear advantages in preserving *local* details, there are some trade-offs to consider. In certain cases, message-passing GNNs, which aggregate information across neighbors, may be preferable, in particular for tasks where *global* context is more important than local details (see Coupette et al. (2025) for a recent analysis of graph-learning datasets under different perspectives). Furthermore, while our method is computationally feasible on medium-sized datasets (as demonstrated in our experiments), the complexity of "naïvely" calculating $\ell$-ECTs increases for larger $k$ and with the size and density of the graph, suggesting a need for improved methods (see Section A.2 for an extended discussion).

## 5. Experiments

In this section, we present experiments to empirically evaluate the performance of the $\ell$-ECT-based approach in graph representation learning, focusing on node-classification tasks. We aim to demonstrate how $\ell$-ECT representations can capture structural information more effectively than traditional message-passing mechanisms, especially in scenarios with high heterophily (even though we consider other scenarios as well). Our experiments compare the performance of $\ell$-ECT-based

models to several standard GNN models, namely graph attention networks (Veličković et al., 2018, GAT), graph convolutional networks (Kipf & Welling, 2017, GCN), graph isomorphism networks (Xu et al., 2019, GIN), as well as a heterophily-specific architecture (Zhu et al., 2020, H2GCN). Furthermore, we showcase how the rotation-invariant metric from Section 4 may be used for spatial alignment of graph data.

### 5.1. $\ell$-ECTs in Graph Representation Learning

The link between message passing and $\ell$-ECTs (cf. Theorem 1) encourages us to empirically validate the expressivity of $\ell$-ECTs for node-classification tasks. Given a featured graph $\mathcal{G}$ and fixed $k \geq 0$, we assign $\ell\text{-}\mathrm{ECT}_k(x; \mathcal{G})$ to every node $v \in \mathcal{G}$. We then use the $\ell$-ECT corresponding to a node together with the respective node feature vector as the *input* to classification models. Subsequently, we focus on XGBoost (Chen & Guestrin, 2016), as we found it to outperform more complex models. However, our $\ell$-ECT can be used with *any* model. Notice that our experiments are not about claiming state-of-the-art performance but rather about showcasing that an approach based on $\ell$-ECT yields results that are *on a par with and often superior* to more complex graph-learning techniques based on message passing, while at the same time working well in *both* heterophilous and homophilous settings.

**Implementation details** We assume that we are given a featured graph $\mathcal{G}$ such that there is an assignment $V(\mathcal{G}) \to \mathcal{Y}$, with $V(\mathcal{G})$ being the node set of $\mathcal{G}$ and $\mathcal{Y}$ being the space of classes w.r.t. the underlying node-classification task. For a fixed $k \geq 0$, $x \in V(\mathcal{G})$ and $N_k(x; \mathcal{G})$ being the $k$-hop neighborhood of $x$ in $\mathcal{G}$, we then approximate $\ell\text{-}\mathrm{ECT}_k(x; \mathcal{G})$ via $\overline{\mathrm{ECT}}(N_k(x; \mathcal{G}))_{(m,l)}$ for sampled directions and filtration steps, as explained in Section 4. We use $m = l = 64$ (but the number of samples may be tuned in practice) and use the the resulting $m \cdot l$-dimensional vector(s) $\overline{\mathrm{ECT}}(N_k(x; \mathcal{G}))_{(m,l)}$, together with the feature vector of $x$, as additional inputs for the classifier. The architecture of our baseline models includes a two-layer MLP after every graph-neighborhood aggregation layer, as well as *skip connections* and *layer normalization*. We train each model for 1000 epochs and report the test accuracy corresponding to the state of the model that admits the *maximum validation accuracy* during training. This makes the predictive performance of our baseline models directly comparable with Platonov et al. (2023).

**WebKB Datasets** For all datasets of the WebKB collection (Pei et al., 2020), our $\ell$-ECT-based approach outperforms the baseline GNNs by far (cf. Table 1; GraphSAGE results from Xu et al. 2024). While the combination of both $\ell\text{-}\mathrm{ECT}_1$ and $\ell\text{-}\mathrm{ECT}_2$ performs best for

Table 1: Performance (*accuracy*, in percent) of graph-learning models on WebKB datasets (5 training runs).

| Model | Cornell | Wisconsin | Texas |
|---|---|---|---|
| GCN | $45.0 \pm 2.2$ | $44.2 \pm 2.6$ | $47.3 \pm 1.5$ |
| GAT | $44.7 \pm 2.9$ | $48.2 \pm 2.0$ | $51.7 \pm 3.2$ |
| GIN | $46.5 \pm 3.1$ | $49.7 \pm 2.5$ | $54.2 \pm 2.9$ |
| GraphSAGE | $\mathbf{76.0 \pm 3.5}$ | $72.9 \pm 1.9$ | $71.8 \pm 2.4$ |
| H2GCN | $66.2 \pm 3.5$ | $70.2 \pm 2.3$ | $72.3 \pm 3.0$ |
| $\ell$-ECT$_1$ | $66.8 \pm 4.2$ | $\mathbf{81.2 \pm 2.9}$ | $74.6 \pm 0.5$ |
| $\ell$-ECT$_2$ | $67.0 \pm 4.9$ | $76.1 \pm 2.8$ | $73.8 \pm 2.6$ |
| $\ell$-ECT$_1$ + $\ell$-ECT$_2$ | $67.1 \pm 4.1$ | $78.5 \pm 2.6$ | $\mathbf{74.8 \pm 3.1}$ |

Table 2: Performance (*accuracy*, in percent) of graph-learning models on *heterophilous* datasets (5 training runs).

| Model | Amazon Ratings | Roman Empire |
|---|---|---|
| GCN | $42.3 \pm 0.7$ | $73.3 \pm 0.8$ |
| GAT | $44.6 \pm 0.9$ | $76.4 \pm 1.2$ |
| GIN | $44.1 \pm 0.8$ | $56.8 \pm 1.0$ |
| GraphSAGE | $42.2 \pm 0.5$ | $77.4 \pm 0.8$ |
| H2GCN | $40.1 \pm 0.7$ | $64.2 \pm 0.9$ |
| $\ell$-ECT$_1$ | $48.4 \pm 0.3$ | $80.4 \pm 0.4$ |
| $\ell$-ECT$_2$ | $49.6 \pm 0.3$ | $78.0 \pm 0.3$ |
| $\ell$-ECT$_1$ + $\ell$-ECT$_2$ | $\mathbf{49.8 \pm 0.3}$ | $\mathbf{81.1 \pm 0.4}$ |

"Texas," using only $\ell$-ECT$_1$ leads to best performance for "Wisconsin." However, for the two aforementioned datasets, the combination of $\ell$-ECT$_1$ and $\ell$-ECT$_2$ only slightly improves the performance in comparison to $\ell$-ECT$_1$, suggesting that 1-hop neighbor information is already sufficiently informative here.

**Heterophilous Datasets** Platonov et al. (2023) introduced several *heterophilous* datasets; we validate our method on "Amazon Ratings" and "Roman Empire," again observing that the combination of $\ell$-ECT$_1$ + $\ell$-ECT$_2$ performs best, substantially outperforming baseline models (cf. Table 2). The results are closely aligned with findings by Platonov et al. (2023), i.e., that specialized architectures like H2GCN often perform less well than "standard" architectures. Moreover, $\ell$-ECT$_1$ outperforms $\ell$-ECT$_2$ on "Roman Empire," while $\ell$-ECT$_2$ outperforms $\ell$-ECT$_1$ on "Amazon Ratings." We interpret this as 1-hop neighborhoods being particularly informative for "Roman Empire," while 2-hop neighborhoods are more informative for "Amazon Ratings."

**Amazon dataset** The Amazon dataset (Shchur et al., 2018) consists of the two co-purchase graphs "Computers" and "Photo." While GAT outperforms all methods on "Computers," the combination of $\ell$-ECT$_1$ and $\ell$-ECT$_2$ performs best on "Photo" (cf. Table 3). Overall, $\ell$-ECT-based methods exhibit competitive performance here, given that they are *not* based on message passing.

Table 3: Performance (*accuracy*, in percent) of graph-learning models on Amazon datasets (5 training runs).

| Model | Computers | Photo |
|---|---|---|
| GCN | $91.6 \pm 1.6$ | $93.6 \pm 1.7$ |
| GAT | $\mathbf{92.4 \pm 1.3}$ | $94.8 \pm 1.1$ |
| GIN | $55.9 \pm 1.5$ | $82.2 \pm 1.3$ |
| GraphSAGE | $89.2 \pm 0.9$ | $92.5 \pm 0.7$ |
| H2GCN | $84.5 \pm 1.4$ | $92.8 \pm 1.2$ |
| $\ell$-ECT$_1$ | $89.6 \pm 0.3$ | $94.1 \pm 0.3$ |
| $\ell$-ECT$_2$ | $90.1 \pm 0.5$ | $94.4 \pm 0.7$ |
| $\ell$-ECT$_1$ + $\ell$-ECT$_2$ | $92.2 \pm 0.6$ | $\mathbf{94.9 \pm 0.6}$ |

Table 4: Performance (*accuracy*, in percent) of graph-learning models on *heterophilous* datasets (5 training runs).

| Model | Actor | Squirrel | Chameleon |
|---|---|---|---|
| GCN | $30.7 \pm 2.1$ | $28.9 \pm 1.4$ | $42.8 \pm 1.8$ |
| GAT | $31.1 \pm 1.8$ | $31.8 \pm 1.3$ | $47.3 \pm 1.3$ |
| GIN | $26.5 \pm 2.0$ | $35.4 \pm 1.5$ | $43.1 \pm 1.7$ |
| GraphSAGE | $30.2 \pm 1.4$ | $33.3 \pm 0.7$ | $45.2 \pm 1.3$ |
| H2GCN | $30.7 \pm 1.9$ | $\mathbf{40.8 \pm 1.4}$ | $\mathbf{62.7 \pm 1.6}$ |
| $\ell$-ECT$_1$ | $\mathbf{31.4 \pm 1.9}$ | $35.6 \pm 0.7$ | $43.5 \pm 1.7$ |
| $\ell$-ECT$_2$ | $30.1 \pm 1.3$ | $35.6 \pm 0.8$ | $40.4 \pm 1.5$ |
| $\ell$-ECT$_1$ + $\ell$-ECT$_2$ | $30.9 \pm 0.7$ | $35.3 \pm 1.5$ | $43.9 \pm 0.7$ |

**Actor/Wikipedia Datasets** Moving to additional heterophilous datasets with high feature dimensionality, we compare predictive performance on "Actor" (Pei et al., 2020) as well as "Chameleon" and "Squirrel" (Rozemberczki et al., 2021); cf. Table 4. For "Actor", the $\ell$-ECT$_1$ model achieves the highest accuracy, while $\ell$-ECT$_1$ + $\ell$-ECT$_2$ performs slightly worse. $\ell$-ECT$_2$ performs the lowest on this dataset, suggesting that larger neighborhoods are detrimental here. For the other datasets, the heterophily-specific model H2GCN performs best. However, $\ell$-ECT$_1$ and $\ell$-ECT$_1$ and $\ell$-ECT$_2$ exhibit similar (or even better) performance as *all other standard baselines*, showing the advantages of $\ell$-ECT methods even in the absence of hyperparameter tuning.

Table 5: Performance (*accuracy*, in percent) of graph-learning models on "Planetoid" datasets (5 training runs).

| Model | Cora | CiteSeer | PubMed |
|---|---|---|---|
| GCN | $88.1 \pm 1.2$ | $74.6 \pm 1.5$ | $85.3 \pm 4.7$ |
| GAT | $\mathbf{88.3 \pm 1.1}$ | $\mathbf{75.3 \pm 1.5}$ | $85.7 \pm 4.2$ |
| GIN | $85.0 \pm 1.5$ | $72.2 \pm 1.7$ | $87.0 \pm 0.5$ |
| GraphSAGE | $82.2 \pm 1.2$ | $68.1 \pm 1.2$ | $84.3 \pm 0.7$ |
| H2GCN | $85.4 \pm 1.6$ | $72.4 \pm 1.9$ | $86.4 \pm 0.5$ |
| $\ell$-ECT$_1$ | $87.6 \pm 0.6$ | $72.1 \pm 0.6$ | $90.2 \pm 0.5$ |
| $\ell$-ECT$_2$ | $87.2 \pm 0.7$ | $72.3 \pm 0.8$ | $\mathbf{90.3 \pm 0.5}$ |
| $\ell$-ECT$_1$ + $\ell$-ECT$_2$ | $87.8 \pm 0.6$ | $72.5 \pm 0.7$ | $\mathbf{90.3 \pm 0.5}$ |

**Planetoid Datasets** We also analyze node-classification performance on datasets from the "Planetoid" collection (Yang et al., 2016), comprising "Cora," "CiteSeer," and "PubMed." We trained all models using a random 75–25 split; cf. Table 5. Although GCN and GAT perform slightly better than $\ell$-ECT methods for "Cora" and "CiteSeer," the gap is surprisingly small. For "PubMed," the $\ell$-ECT-based models even outperform both *all comparison partners*. These findings suggest that the *theoretical* expressivity of $\ell$-ECTs, which we formally established in Section 4, also has *practical* implications, providing an alternative way of dealing with graph data that is not restricted by the underlying model architecture and therefore allows for interpretability.

**Post-hoc Evaluation** To evaluate our methods *across* datasets, we used critical difference diagrams, enabling us to compare the performance of various models on both homophilic and heterophilic graph datasets (cf. Section A.7). The results highlight the superior performance of $\ell$-ECT-based approaches over standard baselines and heterophily-specific architectures such as H2GCN. Notably, the $\ell$-ECT$_1$ + $\ell$-ECT$_2$ method achieved the best average rank of 2 and even the least effective $\ell$-ECT-based model ($\ell$-ECT$_2$) outperformed all non-$\ell$-ECT-based methods, including GAT. Further evaluation against heterophily-specific models reported in the literature corroborates these findings. In comparison to state-of-the-art methods such as H2GCN, GPR-GNN, and GloGNN, $\ell$-ECT$_1$ + $\ell$-ECT$_2$ achieved a competitive rank, matching GloGNN and surpassing both GAT and GT. Despite being a *general-purpose approach* not specifically designed for heterophilic graphs, $\ell$-ECT-based methods demonstrated exceptional adaptability and robustness across diverse graph structures. These results establish $\ell$-ECT-based architectures as a versatile and high-performing solution for node classification tasks, suitable for tackling challenges across a wide range of graph data.

**Summary of Node-Classification Experiments** We find that $\ell$-ECTs work particularly well in situations where aggregating neighboring information is *inappropriate*, such as when dealing with graphs that exhibit a high degree of heterophily. In such contexts, our approach may outperform message-passing-based methods. The upshot of our method is that local graph information can be incorporated *without* the architectural necessity to diffuse information along the graph structure, as it is the case for message-passing-based models. While this discrete diffusion process induced by message passing is useful for a plethora of graph-learning tasks, it can also be an obstruction in learning the right representation for tasks where node features of neighbors in the graph should not be aggregated (cf. Coupette et al. 2025 for a recent analysis of graph-learning datasets in

the context of message passing). In this sense, $\ell$-ECTs naturally overcome a *fundamental limitation* inherent to message-passing methods. Another advantage of $\ell$-ECTs is that they are *agnostic* to the choice of the downstream model. This permits us to use models that are easy to tune, enabling practitioners to make use of their graph data without necessarily having specialized knowledge in GNN training and tuning while at the same time also working well in the small-sample regime. Moreover, it permits using models that are *interpretable*, making our method well-suited for domains where regulatory demands often ask for levels of interpretability that cannot readily be achieved by (graph) neural networks. In fact, by using feature importance values (which are directly available for tree-based algorithms like XGBoost) and since the entries of the $\ell$-ECT vectors that are used as the input for the model can be linked to the directions in the calculation of the $\ell$-ECTs, one may obtain a deeper understanding of *how* the model arrives at predictions (see Section A.3 for a more in-depth discussion and an ablation on the number of directions used to calculate $\ell$-ECTs).

## 5.2. Learning Spatial Alignment of Geometric Graphs

In the following, we use the approach described in Section 4 in order to learn the spatial alignment of two data spaces by re-rotating one into the other. We start by showing that synthetic data, which only differs up to a rotation, can be re-aligned using $\ell$-ECTs. Moreover, we show that this alignment is stable with respect to noise, making it a robust measure for the comparison of local neighborhoods in data. In comparison to other spatial alignment methods like the iterative closest point algorithm, ours does *not* necessitate the computation of all pairwise distances between points in the respective spaces. The latter is often a computational bottleneck, especially for large datasets, thus positioning our method for spatial alignment as a computationally more efficient method in practice. While alignment methods like *Procrustes alignment* are restricted to point-cloud data, we observe that our approach is also capable of aligning embedded graph data (or, more generally, simplicial complexes). This makes it particularly useful for dealing with *geometric graphs*, constituting a highly-efficient alternative to more involved machine-learning models like geometric GNNs (Joshi et al., 2023).

For the subsequent learning problem (see Section A.5 for an example of how to align point-cloud data), we assume that we are given two embedded simplicial complexes $X, Y \subset \mathbb{R}^n$. In light of Section 4, the metric properties of $d_{\text{ECT}}$ ensure that $d_{\text{ECT}}(X, Y) = 0$ if $X$ and $Y$ only differ up to a rotation. We therefore approximate $d_{\text{ECT}}(X, Y)$ via

$$\min_{\rho \in \text{SO}(n)} \left\| \overline{\text{ECT}}(X)_{(m,l)} - \overline{\text{ECT}}(\rho Y)_{(m,l)} \right\|_2^2 \quad (7)$$

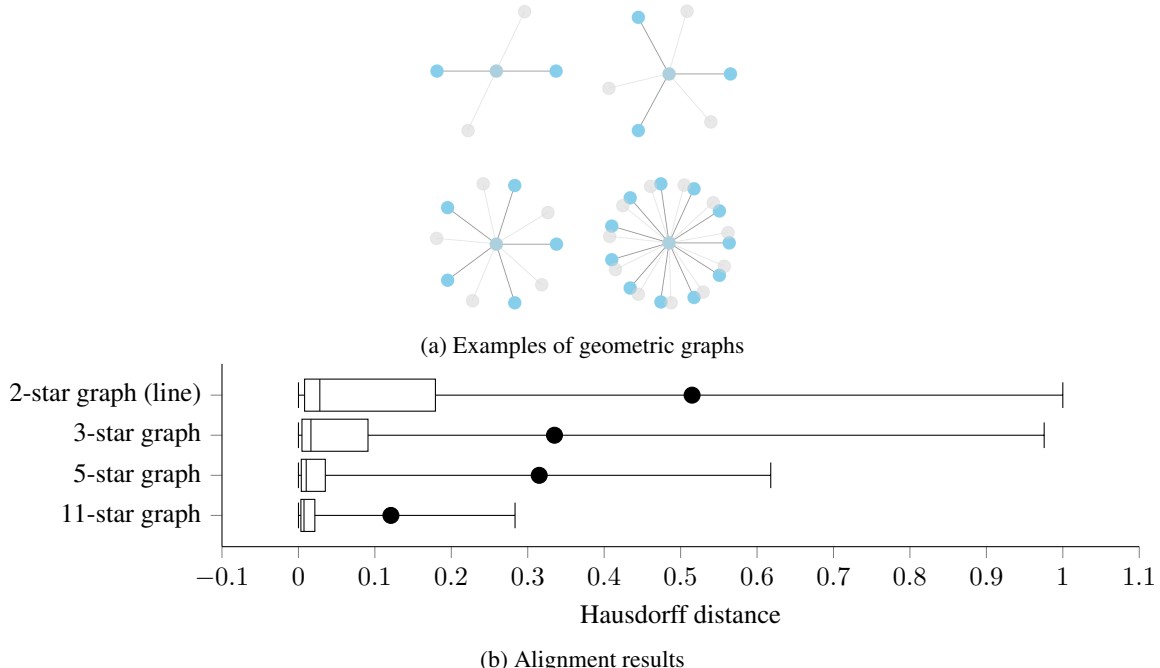

(a) Examples of geometric graphs

(b) Alignment results

Figure 1: A comparison of the Hausdorff distances of aligned graphs. The black dots represents the Hausdorff distance between the original graph and a randomly-rotated version of itself. Our $\ell$-ECT-based alignment always results in substantially lower distances, with a median distance close to zero.

for a choice of directions $v_1, \ldots, v_m \in S^{n-1}$ and filtration steps $t_1, \ldots, t_l \in \mathbb{R}$. As explained in Section 4, the ECT approximations are given by vectors, making it feasible to approach the above learning problem by any gradient-based learning algorithm. The advantage of this formulation is that it yields both the rotation-invariant loss and the rotation that leads to this minimum loss.

Geometric graphs provide a compelling example of the utility of our method, particularly in addressing the challenging problem of graph re-alignment. For this analysis, we focus on a specific type of geometric graph known as the $k$-star graph. A $k$-star graph is defined as a tree with one internal node and $k$ leaves, i.e., a simple structure with relevant geometric properties (cf. Figure 1a). To embed such graphs into a 2D space, we assign a unique 2D vector to each node, ensuring the assigned vectors are equidistant to maintain structural symmetry. Furthermore, to introduce variability and assess robustness, we subject each embedded graph to a random 2D rotation, simulating realistic perturbations encountered in practical settings. The central goal is to recover the original graph's orientation by learning the rotation matrix using the metric defined Section 4. We evaluate the performance of our approach by measuring the similarity between the original graph and its re-rotated version using the *Hausdorff distance*, a metric that quantifies the maximal deviation between two sets of points. To ensure significance, we repeated the learning procedure

200 times, maintaining consistent initializations for both the graph and the rotation matrix. Figure 1 shows the results; we observe that our realignment procedure *consistently* achieves small Hausdorff distances with medians near zero, indicating the successful recovery of the original graph's orientation. By contrast, the Hausdorff distances between the original graph and its perturbed version are significantly larger. While equivariant GNNs have been shown to struggle with distinguishing the orientation of rotationally-symmetric structures (Joshi et al., 2023), our method generates graph representations that are initially sensitive to rotations but can be made rotation-invariant through alignment of the underlying ECTs.

These findings highlight the potential of ECT-based metrics for *robust* geometric graph alignment, paving the way for broader applications in domains requiring precise graph-based comparisons such as the analysis of geometric graphs with constrained parameter budgets (Maggs et al., 2024). This result is particularly notable since the $\ell$-ECT easily outperforms more involved architectures, pointing towards its overall utility as an alternative to message-passing graph neural networks. At the same time, we believe that the $\ell$-ECT could also help in aligning higher-order data like geometric simplicial complexes.

## 6. Discussion

We introduced the *Local Euler Characteristic Transform* ($\ell$-ECT), providing a novel approach to graph-representation learning that preserves local structural information *without* relying on aggregation. Our method addresses fundamental limitations in message-passing neural networks, particularly in tasks where aggregating neighboring information is suboptimal, such as in graphs with heterophily. By retaining critical local details, $\ell$-ECTs enable more nuanced and expressive representations, offering significant advantages in node classification tasks. One key strength of our approach is its model-agnostic nature, allowing it to be paired with interpretable machine learning models (in our experiments, an XGBoost model was used to provide feature importance values, for instance). This is particularly useful in domains such as healthcare, finance, and legal applications, where regulatory frameworks demand high levels of transparency and interpretability that are often difficult to achieve with black-box neural networks. By leveraging $\ell$-ECTs, we can satisfy these requirements while maintaining the high expressivity and high predictive performance required for graph-learning tasks. In this way, $\ell$-ECT-based representations offer a novel pathway toward interpretable machine learning on graph data: they yield topologically-grounded, vectorized encodings of local structure that not only retain predictive power but also support downstream diagnostics.

**Future Work: Higher-Order Domains** Being situated at the intersection of geometry and topology, our $\ell$-ECT method is part of the nascent field of *topological deep learning* (Papamarkou et al., 2024), which aims to develop novel inductive biases that are capable of leveraging additional structural information from data, both in the context of graphs and in the context of higher-order domains like *simplicial complexes*. It is in this context where we believe that future work could be beneficial, in particular since recent research (Ballester et al., 2025) showed that tasks on such domains are highly challenging for existing GNNs. Given the advantageous scalability properties of the ECT (Röell & Rieck, 2024; Turner et al., 2014), we believe that this constitutes a useful avenue for future research.

**Future Work: Comparing Representations** Containing both geometrical and topological components, we also believe ECT-based methods to be suitable in navigating different representations. Since the ECT can be considered a compression technique with controllable fidelity properties (Röell & Rieck, 2025), it could be useful in condensing latent spaces, thus permitting simple and efficient comparisons of models as hyperparameters are being varied. Such *multiverse analyses* are vital for ensuring reproducibility (Bell et al., 2022; Wayland et al., 2024).

**Future Work: Hybrid Models** Beyond representation learning on graphs, our $\ell$-ECT framework also opens up new applications in domains where local structure is critical, such point-cloud analysis (including sensor data or computer-graphics data), 3D shape analysis, or data from the life sciences (like molecular data or biological networks). Future work could thus explore more efficient algorithms for computing ECTs and $\ell$-ECTs at scale, as well as hybrid approaches that balance local and global information more effectively. A highly-relevant direction would be the integration of $\ell$-ECTs into existing message-passing neural networks, similar to recent work that leverages persistent homology (Verma et al., 2024). Moreover, heterophily-specific mechanisms such as a separation of neighborhood aggregation (as used in specialized GNN architectures) may be incorporated into our $\ell$-ECT-based framework to further strengthen its expressivity in the presence of high-heterophily graphs.

## Software and Data

Our code is available under `https://github.com/aidos-lab/Diss-l-ECT`. We make use of standard benchmarking datasets, loaded and processed via the `PyTorch Geometric` library (Fey & Lenssen, 2019).

## Acknowledgments

The first author acknowledges the use of ChatGPT for grammar suggestions. Both authors are very grateful for the discussions with the anonymous reviewers, in particular reviewer `nYTM`, and the area chair, who also believed in the merits of this work. This work has received funding from the Swiss State Secretariat for Education, Research, and Innovation (SERI).

## Impact Statement

This paper presents work whose goal is to advance the field of graph machine learning. There are many potential societal consequences of our work, none which we feel must be specifically highlighted here.

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

# A. Appendix

## A.1. Proofs

We briefly restate all theorem from the main text for the reader's convenience before providing proofs.

**Theorem 1.** *Let $\mathcal{G}$ be a featured graph and let $\{\ell\text{-ECT}_1(x;\mathcal{G})\}_x$ be the collection of local ECTs with respect to the 1-hop neighborhoods in $\mathcal{G}$. Then the collection $\{\ell\text{-ECT}_1(x;\mathcal{G})\}_x$ provides the necessary (non-learnable) information for performing a single message-passing step on $\mathcal{G}$, in the sense that for a given vertex $x \in \mathcal{G}$ one can reconstruct the feature vectors of its 1-hop neighborhood from $\ell\text{-ECT}_1(x;\mathcal{G})$.*

*Proof.* By the remark in Section 4 (in the paragraph right above the original statement of this Theorem), we may assume that the natural embedding of $\mathcal{G}$ into $\mathbb{R}^n$ is a graph isomorphism. Then, making use of the invertibility theorem, the 1-hop neighborhood of a point $x$ in the embedding of $\mathcal{G}$ can be reconstructed from $\ell\text{-ECT}_1(x;\mathcal{G})$. Therefore, the feature vectors of $x$ and its 1-hop neighbors can be deduced from $\ell\text{-ECT}_1(x;\mathcal{G})$, which is the only non-learnable information one needs to perform a message-passing step. $\qquad\square$

**Theorem 2.** *Two featured graphs $\mathcal{G}_1$ and $\mathcal{G}_2$ are isomorphic if and only if $\text{ECT}(\mathcal{G}_1) = \text{ECT}(\mathcal{G}_2)$.*

*Proof.* When two featured graphs are isomorphic in the sense of Definition 2, their respective Euclidean embeddings produce equal ECTs by construction because the node feature vectors of two corresponding points under the isomorphism are equal. By contrast, let us assume that $\text{ECT}(\mathcal{G}_1) = \text{ECT}(\mathcal{G}_2)$. Then by the invertibility theorem, the Euclidean embeddings of $\mathcal{G}_1$ and $\mathcal{G}_2$ are equal. Therefore, the only information that may tell apart the two graphs are their node labels, but this means that $\mathcal{G}_1$ and $\mathcal{G}_2$ are isomorphic. $\qquad\square$

**Theorem 3.** *$d_{\text{ECT}}$ is a metric on the collection of rotation classes of finite simplicial complexes embedded in $\mathbb{R}^n$.*

*Proof.* $d_{\text{ECT}}(X, X) = 0$ holds for $\rho$ being the identity. Now assume that $d_{\text{ECT}}(X, Y) = 0$. Then there exists $\rho \in \text{SO}(n)$ with $\|(\text{ECT}(X) - \text{ECT}(\rho Y))\|_\infty = 0$. As $\|\bullet\|_\infty$ is a norm, it follows that $\text{ECT}(X) = \text{ECT}(\rho Y)$, and by the invertibility theorem we obtain $X = \rho Y$. This shows the first property of a metric (note that positivity follows from $\|\bullet\|_\infty$). For symmetry, note that $\|(\text{ECT}(X) - \text{ECT}(\rho Y))\|_\infty = \|(\text{ECT}(\rho^{-1}X) - \text{ECT}(Y))\|_\infty$ since rotations are invertible. For the triangle inequality, let $Z$ be another finite simplicial complex. $d_{\text{ECT}}(X, Z)$ then reads $\inf_{\rho \in \text{SO}(n)} \|(\text{ECT}(X) - \text{ECT}(\rho Z))\|_\infty$, which is less than or equal to $\inf_{\rho, \rho' \in \text{SO}(n)}(\|(\text{ECT}(X) - \text{ECT}(\rho' Y))\|_\infty + \|(\text{ECT}(\rho' Y) - \text{ECT}(\rho Z))\|_\infty)$. This term, however, is equal to $\inf_{\rho, \rho' \in \text{SO}(n)}(\|(\text{ECT}(X) - \text{ECT}(\rho' Y))\|_\infty + \|(\text{ECT}(Y) - \text{ECT}((\rho')^{-1}\rho Z))\|_\infty)$, which is equal to $\inf_{\rho \in SO(n)} \|(\text{ECT}(X) - \text{ECT}(\rho Y))\|_\infty + \inf_{\rho \in \text{SO}(n)} \|(\text{ECT}(Y) - \text{ECT}(\rho Z))\|_\infty$. But this final term is *precisely* the definition of $d_{\text{ECT}}(X, Y) + d_{\text{ECT}}(Y, Z)$. $\qquad\square$

## A.2. Computational Complexity

For a fixed node $x$, the computational complexity of $\ell\text{-ECT}_k(x)$ is $O(m \cdot l \cdot |N_k(x)|)$, where: (i) $m$ is the number of sampled directions, (ii) $l$ is the number of filtration steps, and (iii) $|N_k(x)|$ is the number of vertices (or simplices) in the $k$-hop neighborhood of $x$.

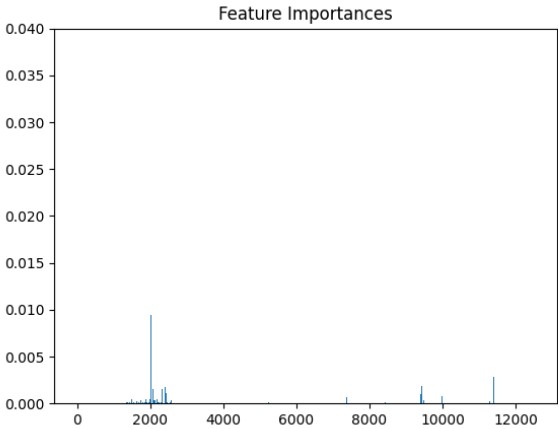

Figure 2: Feature importance scores of an XGBoost model for the "Coauthor Physics" dataset (using $\ell$-$\mathrm{ECT}_1$). Only a small number of features admit high importance scores.

### A.3. Ablation on Directions and Interpretability

Coming back to our approximation of $\mathrm{ECT}(X)$ via $\overline{\mathrm{ECT}}(X)_{(m,l)} := \mathrm{ECT}(X)_{|\{v_1,\ldots,v_m\}\times\{t_1,\ldots,t_l\}}$ for uniformly-distributed directions $v_1,\ldots,v_m \in S^{n-1}$ and filtration steps $t_1,\ldots,t_l \in \mathbb{R}$, we notice that the $(l \cdot (j-1)+1)$-th till $(l \cdot j)$-th entries of $\overline{\mathrm{ECT}}(X)_{(m,l)}$ correspond to the direction $v_j$. The latter gives us the opportunity to get a deeper understanding of how the model predicts its outcome, by analyzing its feature importance values (which are available for tree-based algorithms like XGBoost). Therefore, our approach enables us to analyze which features, i.e., directions, of the underlying ECT vector are most important. In practice, we often observe that a small number of features admits high feature importance with respect to the corresponding model (cf. Figure 2). This raises the question if we may use a *smaller* random collection of features and still obtain reasonably useful results. We therefore ran experiments for a collection of datasets for a varying number of randomly-sampled entries of the $\ell$-$\mathrm{ECT}_1$ vector; cf. Table 6. Here, 4096 corresponds to the whole vector. We observe that for certain datasets, such as "Coauthor CS," "Coauthor Physics," and "Amazon Ratings," the performance of the model only slightly changes when using a reduced version of the $\ell$-$\mathrm{ECT}_1$ vector. In light of the results by Curry et al. (2022), this observation is not entirely surprising—one main claim therein is that the ECT can be determined using a small number of directions.

Table 6: Mean accuracy (in percent, 5 runs each) for different node-classification tasks, and varying numbers of randomly-sampled entries of the corresponding $\ell$-$\mathrm{ECT}_1$ vectors.

| Dataset | 0 | 50 | 100 | 500 | 1000 | 4096 |
|---|---|---|---|---|---|---|
| WikiCS | 67.8 | 69.2 | 70.5 | 71.3 | 72.7 | 74.6 |
| Coauthor CS | 92.1 | 92.3 | 92.4 | 92.5 | 92.6 | 92.6 |
| Coauthor Physics | 95.2 | 95.6 | 95.6 | 95.8 | 95.9 | 96.1 |
| Roman Empire | 64.7 | 73.7 | 75.8 | 78.3 | 79.7 | 80.4 |
| Amazon Ratings | 47.9 | 47.9 | 48.2 | 48.4 | 48.2 | 48.4 |

Table 7: Performance (*accuracy*, in percent) of graph-learning models on WikiCS dataset (5 training splits).

| Model | WikiCS |
|---|---|
| GCN | $75.2 \pm 0.8$ |
| GAT | $78.7 \pm 1.2$ |
| GIN | $74.2 \pm 1.7$ |
| GraphSAGE | $73.4 \pm 1.5$ |
| H2GCN | $75.3 \pm 1.4$ |
| $\ell\text{-ECT}_1$ | $74.6 \pm 0.5$ |

Figure 3: A comparison of the squared $L^2$ distances of $\ell$-ECTs of aligned and non-aligned MNIST digits of "1," respectively.

### A.4. Additional Node Classification Experiment

**WikiCS Dataset**  To further validate the effectiveness of our approach, we consider the WikiCS dataset (Mernyei & Cangea, 2020), a medium-sized co-occurrence graph derived from Wikipedia articles on computer science topics. Nodes represent articles, and edges reflect mutual links between them. Each node is equipped with a 300-dimensional embedding, and the task is to classify articles into one of several predefined categories. As shown in Table 7, our $\ell\text{-ECT}_1$-based method achieves competitive performance compared to message-passing baselines. While GAT obtains the best accuracy overall, $\ell\text{-ECT}_1$ performs on par with H2GCN and GCN, despite not relying on neighborhood aggregation. This supports the idea that $\ell$-ECT-based representations can serve as effective input features in classification settings, even for graphs with moderately homophilous structures. The small standard deviation further illustrates the stability of our method across splits.

### A.5. Spatial Alignment of High-Dimensional Data

Following our previous observations that $d_{\text{ECT}}$ enables us to align two spaces, we now use it to investigate its effect on high-dimensional data. We start this discussion with the well-known MNIST benchmark dataset, following an analysis of local geometrical-topological structures that we performed previously (von Rohrscheidt & Rieck, 2023). We thus first represent each (gray-scale) image in the dataset as a 784-dimensional vector, by flattening the image. In this way, we obtain a high-dimensional point cloud corresponding to the dataset. Subsequently, we sample 300 points of digits of "1" and calculate the pairwise distances of their respective $\ell$-ECT (with respect to the whole point cloud), for $k = 10$. Finally, we calculate the pairwise distances of the respective aligned $\ell$-ECTs (by using the approach of Eq. (7) with $k = 10$). Figure 3 shows the results; we observe that the aligned $\ell$-ECTs have a significantly lower squared $L^2$ distance (with a median of $\approx 112$) than the non-aligned ones (with a median of $\approx 224$), showcasing that rotations cause dissimilarity between small neighborhoods of points, in many cases.

### A.6. Homophily Scores of Node-Classification Experiments

Table 8 reveals that our benchmark suite spans the entire range from extreme homophily to extreme heterophily. At the homophilic end lie the Amazon co-purchase graphs "Computers" and "Photo," together with the "Planetoid" citation graphs "Cora," "CiteSeer," and "PubMed." In each of these networks, at least *seven of every ten* edges connect nodes with identical class labels, replicating the conditions under which early message-passing GNNs achieved their seminal successes. Near the middle of the spectrum, "Cornell" and the "Amazon Ratings" datasets exhibit mixed behavior, with roughly one third of their edges being heterophilous. Thus, neighborhood aggregation still conveys useful class-specific information, but the signal is noticeably diluted. The lower end is populated by the remaining datasets, where fewer than *one edge in three* is homophilic, and by "Roman Empire," the most extreme case in our experimental suite, where only about *one edge in twenty* links same-label endpoints.

Table 8: Edge-homophily ratios for every dataset used in our node-classification experiments, sorted in ascending order. As the table shows, our experiments comprise a wide variety of datasets.

| Dataset | $H_{\textbf{edge}}$ |
|---|---|
| Roman Empire | 0.047 |
| Texas | 0.110 |
| Wisconsin | 0.210 |
| Actor | 0.217 |
| Squirrel | 0.220 |
| Chameleon | 0.230 |
| Cornell | 0.300 |
| Amazon Ratings | 0.380 |
| CiteSeer | 0.736 |
| Amazon Computers | 0.777 |
| PubMed | 0.802 |
| Cora | 0.810 |
| Amazon Photo | 0.827 |

Because six datasets are heterophilic, five are strongly homophilic, and two occupy the transition zone, we believe our experimental suite to be effectively *balanced*. A model must therefore operate reliably across sharply different structural regimes to achieve consistently high average rank. Classical message-passing architectures depend on *homophily* and tend to deteriorate as the ratio falls, whereas our empirical results from Section 5 demonstrate that our proposed $\ell$-ECT representations retain competitive—and often even *superior*—accuracy *regardless of homophily level*. The most conspicuous gains appear precisely on the graphs where neighbor aggregation is *least informative*, namely "Roman Empire," "Texas," and "Wisconsin," confirming that $\ell$-ECT features capture structural cues that message passing alone fails to exploit. Hence, the numerical landscape mapped out in Table 8 substantiates the claim that our experimental design both stresses the limits of common GNNs while at the same time showcasing the robustness of $\ell$-ECT-based approaches in settings where label agreement along edges is sparse.

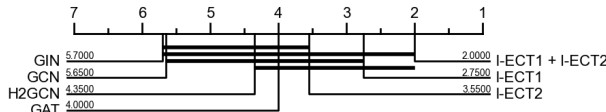

Figure 4: *Critical difference diagram* showing the ranks of different models across all node-classification tasks from Section 5. Even the worst-performing $\ell$-$\mathrm{ECT}$-based approach ($\ell$-$\mathrm{ECT}_2$) exhibits superior performance to *all other methods*, when averaged across all tasks.

### A.7. Post-hoc Evaluation of Node-Classification Experiments

A critical difference diagram arranges the average ranks of multiple models across a set of datasets in order to facilitate overall performance comparisons between the model performances. Such diagrams are commonly used when comparing a suite of models on different datasets (cf. Borgwardt et al. (2020) for similar plots in the context of *graph kernels*).

Figure 4 shows the results for all node-classification results from Section 4, including both homophilic and heterophilic graph datasets. We observe that the $\ell$-$\mathrm{ECT}$-based approaches outperform standard methods and the heterophily-specific architecture H2GCN by far, when averaged over all datasets.[3] The best-performing method $\ell$-$\mathrm{ECT}_1 + \ell$-$\mathrm{ECT}_2$ exhibits an average rank of 2, while the worst performing method is GIN with an average rank of 5.7. Even the worst-performing $\ell$-$\mathrm{ECT}$-based method ($\ell$-$\mathrm{ECT}_2$) performs better than the best non-$\ell$-$\mathrm{ECT}$-based method, i.e., GAT. However, the most interesting fact that can be gleaned from the diagram involves the statistical significance of the results. Methods connected over the same bar are not performing statistically significantly differently. This seemingly *negative* result has a positive implication: Despite being orders of magnitude more complex, even specialized graph neural networks do not perform statistically significantly better than $\ell$-$\mathrm{ECT}$-based methods. Given that our results are based on a standard XGBoost model without *any* task-specific hyperparameter tuning, we believe that this demonstrates the potential and practical utility of our proposed methods.

Table 9: Ranks (lower is better) of models from Platonov et al. (2023) across the *heterophilic* datasets therein, in comparison to our methods. Notice that our method is a *general-purpose* method for node classification and neither geared towards heterophily nor homophily.

| Model | Rank |
| --- | --- |
| H2GCN (Zhu et al., 2020) | 18.3 |
| CPGNN (Zhu et al., 2021) | 16.8 |
| GPR-GNN (Chien et al., 2021) | 15.3 |
| ResNet (He et al., 2016) | 13.8 |
| l-ECT1 | 12.4 |
| l-ECT2 | 12.4 |
| GAT (Veličković et al., 2018) | 12.3 |
| GT (Shi et al., 2021) | 11.0 |
| **l-ECT1 + l-ECT (ours)** | **11.0** |
| GloGNN (Li et al., 2022) | 11.0 |
| ResNet+SGC (Wu et al., 2019) | 10.8 |
| FAGCN (Bo et al., 2021) | 10.0 |
| JacobiConv (Wang & Zhang, 2022) | 9.8 |
| GCN (Kipf & Welling, 2017) | 9.6 |
| GBK-GNN (Du et al., 2022) | 9.0 |
| ResNet+adj (Zheleva & Getoor, 2009) | 7.3 |
| SAGE (Hamilton et al., 2017) | 5.9 |
| GAT-sep (Veličković et al., 2018) | 5.5 |
| GT-sep (Shi et al., 2021) | 5.3 |
| FSGNN (Maurya et al., 2021) | 3.0 |

To further evaluate the performance of our methods in comparison to those reported in the literature, we also included a comparison with the results presented by Platonov et al. (2023), using the ranks of the respective models as the basis for evaluation; cf. Table 9. Among the listed methods, several, such as H2GCN, CPGNN, and GPR-GNN, are explicitly designed for heterophilic graph settings, leveraging specialized architectures to handle the challenges posed by such data. In contrast, our $\ell$-$\mathrm{ECT}_1 + \ell$-$\mathrm{ECT}_2$ method, despite being a general-purpose approach not tailored specifically for heterophilic settings, achieves a competitive rank of 11. This performance is on a par with other top-performing heterophily-specific models, such as GloGNN, and outperforms well-established architectures like GT and GAT by a significant margin. Overall, these results highlight the robustness and adaptability of our method, demonstrating its ability to handle diverse graph structures effectively without requiring customization for heterophilic scenarios. In consideration of the results given in Figure 4, this makes $\ell$-$\mathrm{ECT}$-based approaches a versatile general-purpose solution for node -classification tasks.

---

[3] We used https://github.com/hfawaz/cd-diagram for the creation of the critical difference diagram.

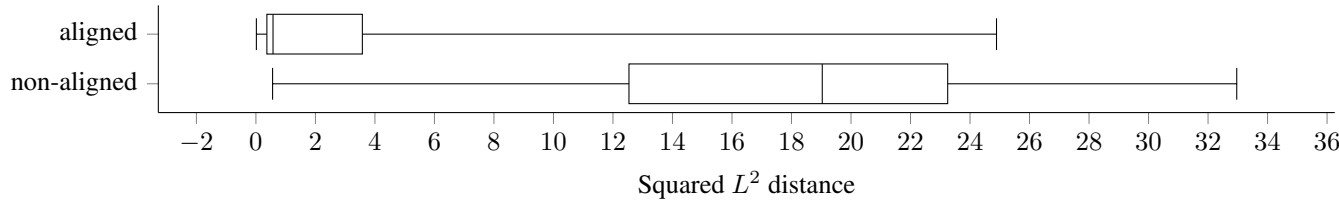

Figure 5: A comparison of the squared $L^2$ distances of the ECTs of aligned and non-aligned wedged spheres, respectively. We see that alignment results in a median loss of zero, thus effectively showing that the two spaces are the same.

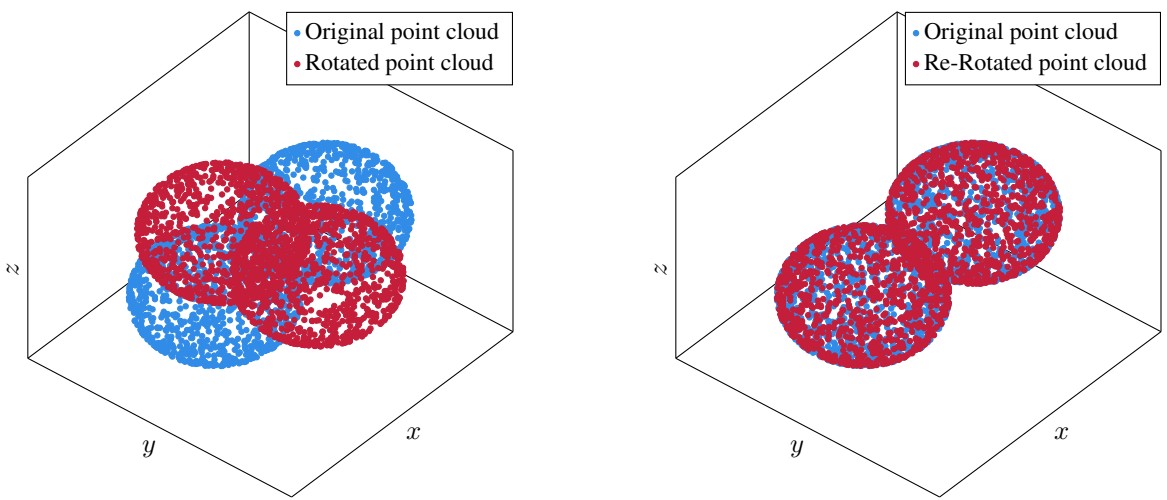

Figure 6: A comparison of two wedged spheres with one being rotated around the wedge point and the points being perturbed by Gaussian noise (left) and the learned re-rotated sphere that is aligned with the original data (right).

### A.8. Spatial Alignment of Wedged Spheres

We approximate the optimization problem from Eq. (7) to show that we can learn a spatial alignment of two data spaces, while the distance between ECTs of non-aligned spaces that only differ up to a rotation will generally be high. We start with a so-called *wedged sphere*, meaning two 2-dimensional spheres which are concatenated at a gluing point (cf. Figure 6 and von Rohrscheidt & Rieck 2023). We use 2000 uniformly-sampled points from such a wedged sphere, and compare the squared $L^2$ loss between the ECTs of this sample and a rotation of the same data space. We repeat this procedure 500 times, where at each step both the sample of the wedged sphere and the rotation matrix which yields the rotated version of the same space are sampled randomly. We notice that the $L^2$ losses between the non-aligned spaces are high (with a median of around 19), whereas the $L^2$ losses of the non-aligned spaces are significantly lower, with a median loss close to zero (cf. Figure 5). Moreover, we observe that the ECT of the same space significantly changes when the coordinate system is transformed, which corroborates the necessity of a rotation-invariant metric for the comparison of ECTs. We conclude that an alignment of the ECTs of the two underlying data spaces in fact leads to an alignment of the data spaces itself, as promised by the theoretical results in Section 4.

**Robustness**    Figure 7 and Figure 8 show that the spatial alignment of wedged spheres still works satisfactorily, even in the presence of outliers and noise. This property is an important feature when dealing with real-world data, which is often noisy, and enables us to align spaces that only *approximately* differ up to a rotation. By contrast, the Hausdorff distance, i.e., a widely-used metric between point clouds is (by definition) *highly sensitive* to outliers. We therefore conclude that the proposed metric based on ECTs is a robust metric to compare point clouds of potentially different cardinalities.

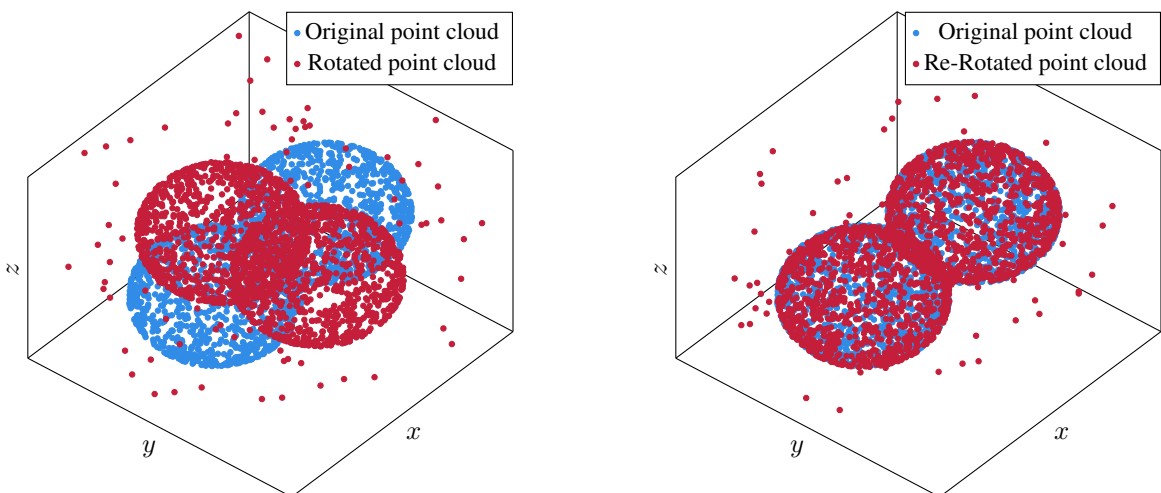

Figure 7: A comparison of two wedged spheres, with one being rotated around the wedge point and added 200 outliers (left) and the learned re-rotated sphere that is aligned with the original data (right).

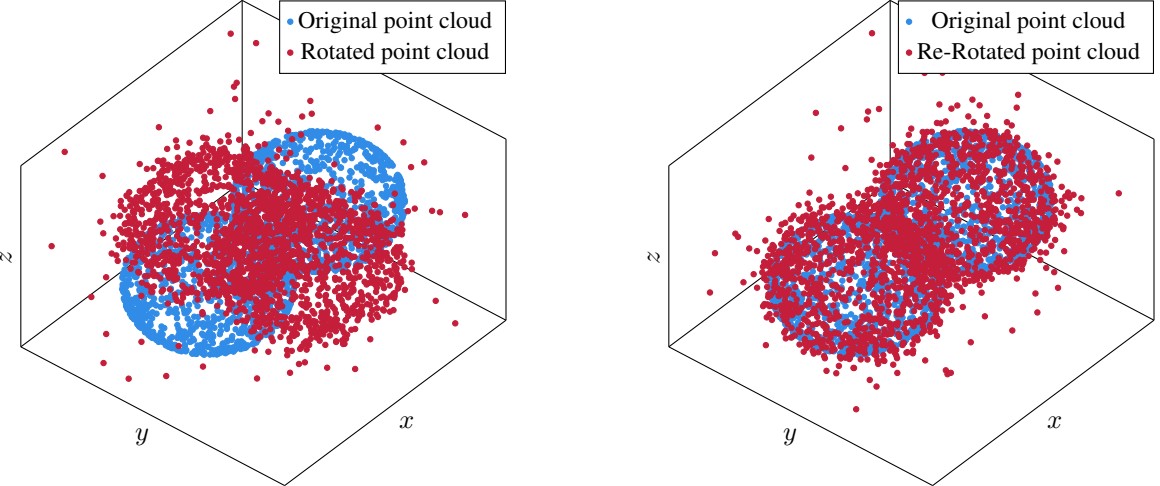

Figure 8: A comparison of two wedged spheres, with one being rotated around the wedge point and the points being perturbed by *Gaussian noise* (left) and the learned re-rotated sphere that is aligned with the original data (right).

