# OpenReview forum: "Diss-l-ECT: Dissecting Graph Data with Local Euler Characteristic Transforms"
_ICML.cc/2025/Conference — ICML 2025 poster_

### Official Review · Reviewer_DNCh · 2025-03-06

**Overall Recommendation:** 3

**Summary:**

In their paper "Diss-l-ECT: dissecting graph data with local Euler characteristic trasnform", the authors suggest a local version of Euler characteristic transform (ECT) that, given a graph with node features, assigns to each node an additional feature vector containing Euler characteristics of local subgraphs with nodes selected by a range of feature-based thresholds. The authors then show that XGBoost on the concatenation of original features and l-ECT features performs very well in various node clasification tasks, in particular for heterophilic graphs.

**Claims And Evidence:**

One important baseline is missing.

**Essential References Not Discussed:**

Not that I know of

**Experimental Designs Or Analyses:**

Seems sensible

**Methods And Evaluation Criteria:**

Yes

**Other Comments Or Suggestions:**

MINOR COMMENTS

* On page 2, simplicial complex is defined abstractly, without any node features. Then page 3 starts with simplicial complex X \in R^n. But what does it mean for X to be in R^n? This was never defined. Featured graphs and feature vectors are only introduced later. This makes it confusing for the reader.

* Several times the authors refer to "point clouds", even calling them an "important special case". I don't understand how l-ECTs would be applicable to point clouds, where the (local) Euler characteristic is just the number of points. This seems to carry no useful information. Please clarify what is meant.

* In section 5.1, for all graphs please specify where they are heterophilic or homophilic (e.g. WebKB datasets are homo- or hetero- philic? it's not stated). Perhaps use some measure of homophily and report the value for each graph.

* "Amazon dataset" paragraph: "combintaion of l-ECT1 and l-ECT2 outperforms on Photo" -- this statement is misleading, the difference is 0.1% with errors being ~1%, clearly not significant. In fact, I think GAT should be bold for Photo in Table 3, because it's not meaningfully different from the top value in the Photo column. I suggest to use bold in all such cases in all tables (e.g. l-ECT2 for Amazon Ratings in Table 2 etc.)

* "Planetoid datasets" -- what is "planetoid"?

* I did not understand Table 7. Does it take ranks from Platonov et al.? If yes, how are your methods added there? Also, is H2GCN the best or the worse? It has the highest rank value, meaning it's the worst? As in Figure 4 (higher value => worse). If FSGNN is the best, why did not you use it in your benchmarking? If H2GCN is the best, then why did you find it so much worse then l-ECT in Figure 4? Very confusing.

**Other Strengths And Weaknesses:**

Strengths: The paper is very clearly written. l-ECT definition makes sense. Experimental results are convincing.

Weaknesses: No intuition about why/when l-ECT works. No baseline comparison with XGBoost on the node features.

Overall, borderline paper but I tend to acceptance.


MAJOR COMMENTS

* The main argument of the paper is that local Euler characteristics can contain information useful for node classification. What would be very helpful is to have some illustration of how this can be the case. Can you show an example of a small toy graph with nodes from two classes and some node features, where local Euler characteristic in 1-hop neighborhood predicts the class? Currently, the paper provides very little intuition for why/when this should work. The authors emphasize heterophilic graphs, so it would be great if the toy graph is heterophilic. For a graph, Euler characteristic is simply the number of nodes minus the number of edges. Maybe the toy graph can have 2D node features and a particular 2D direction and a particular threshold value that would make 1-dimensional l-ECT (m=l=1) perfectly predictive of class? Something along the lines.

* In every table I am missing one IMHO crucial comparison: XGBoost performance on node features without any l-ECT features. Please add this to Tables 1--5 as additional row. Currently it is impossible to say if l-ECT performance is due to XGBoost on original node features performing well or due to l-ECT features actually providing some additional information. If for some of the graphs the performance of XGBoost on original features is as good as for l-ECT, it would mean that l-ECT features are useless for that graph.

* I did not understand the point of section 5.2. Randomly rotated graphs shown in Figure 1 could be aligned using a trivial application of Procrustes rotation. Why does one need l-ECT machinery for that? Is this whole section supposed to be a sanity check that l-ECT alignment works correctly? If so, it should be presented as a "merely" sanity check, and not as a important result that takes 1 page of text and 1 figure. The result seems trivial.

**Questions For Authors:**

See above

**Relation To Broader Scientific Literature:**

Not sure what one is supposed to say here. This paper suggests l-ECM, clearly building up on ECM etc.

**Theoretical Claims:**

Didn't check

---

> ### Author Rebuttal · Authors · 2025-03-28
>
> Dear Reviewer,
>
> we thank you for the constructive and thoughtful feedback. We’re especially grateful for recognizing the clarity of our exposition, the soundness of our method, and the convincing experimental results. Below, we respond to all concerns.
>
> > Question regarding illustration and intuition
>
> Thank you for the helpful suggestion. To clarify, we are not computing the local Euler characteristic, but rather the local Euler Characteristic Transform (l-ECT). Unlike the raw Euler characteristic, l-ECT scans the local neighborhood in multiple directions, computing Euler characteristics of sublevel sets. This directional process, known to be invertible under mild assumptions, provides a rich fingerprint of local geometry and topology. While the Euler characteristic is a topological invariant, the collection of directional profiles captures detailed geometric structure. **We'll add more explanations and illustrations in the revision.**
>
> > “No intuition about why/when l-ECT works. No baseline comparison with XGBoost on the node features.”
>
> Regarding intuition, see the response above; we also plan on illustrating the computation better. Regarding the baseline, we’re happy to include an **additional comparison against XGBoost** for the respective node classification tasks.
>
> > Request for XGBoost on raw node features
>
> We agree that this baseline is important and are happy to **include the corresponding results in our revision.** In the meantime, Table 6 already shows that l-ECT features are informative: as the number of directions used in the transform is reduced, classification performance often degrades substantially. This suggests that the l-ECT encodes nontrivial structural information not already present in the raw features, and that this information contributes meaningfully to the model’s performance.
>
> > Question about spatial alignment using l-ECT
>
> Thank you for raising this. While the experiment in Section 5.2 does serve as a sanity check, it also demonstrates that l-ECT provides a scalable alternative to alignment methods like Procrustes, which rely on costly SVD. Our approach scales linearly with n, l, and m, and allows control over approximation quality via m and l. **We'll revise the text to better explain this motivation and highlight the practical benefits of l-ECT-based alignment.**
>
>
> > Question on the definition of simplicial complexes
>
> You're right—thanks for catching this. We meant geometric, not abstract, simplicial complexes, and **will revise the text to reflect this consistently.**
>
> > Question on l-ECT and point clouds
>
> To clarify, we don’t compute a single Euler characteristic, but use the local Euler Characteristic Transform (l-ECT), which builds a vector from directional filtrations of the local neighborhood. Point clouds are treated as edge-free geometric graphs (0D complexes in R^n. Since the ECT is invertible, it serves as a rich fingerprint of local structure (in this case the point cloud). **We’ll revise the text to make this clearer.**
>
> > Request of homophily score reporting
>
> Thank you for the suggestion. We will add a standard homophily measure (edge homophily ratio) for each dataset and indicate whether each dataset is homophilic or heterophilic. For example, the WebKB datasets exhibit low edge homophily (~0.2) and are thus considered heterophilic, while citation graphs like Cora are homophilic (e.g., ~0.8).
>
> > Request for using bold for all results within error margins
>
> Thank you for pointing this out. **We will update all result tables to use boldface for all values within one standard error of the best performance.** We believe this makes the comparisons clearer and more meaningful.
>
> > “"Planetoid datasets" -- what is "planetoid"?”
>
> “Planetoid” refers to the data introduced in the benchmark (Yang et al., 2016), containing the citation graph datasets Cora, Citeseer, and Pubmed. The dataset is introduced in l.353 of our manuscript. **We will clarify this.**
>
> > Questions about Table 7
>
> Thank you for pointing this out. Table 7 uses the ranking setup from Platonov et al. (2023), with lower mean ranks indicating better performance. We added l-ECT post hoc using the same protocol. FSGNN, while strong, was excluded from our benchmarking due to its specialized design. As the comparison focuses on heterophilic graphs, such methods are favored—but our goal is to show that l-ECT performs well without architectural tuning, **which we’ll clarify in the revision.**
>
> **We are grateful for your recognition of our contributions and your close reading of the manuscript.** If the improvements we propose address your concerns, we would be very grateful if you would consider raising your score.
> Thank you once again for your constructive and thoughtful review, we are happy to address any other questions you might have!
>
> Best regards,
>
>  The Authors

---

> > ### Comment · Reviewer_DNCh · 2025-04-02
> >
> > Thank you for your response. I can see that all 4 reviewers gave the identical score of 3, so the paper has a good chance of getting in. I also have to say that I was a bit surprised that the authors did not do any single new experiment for their rebuttals... Personally I would like to keep the score at 3 -- I continue to recommend acceptance, but only weakly.
> >
> > > To clarify, we are not computing the local Euler characteristic, but rather the local Euler Characteristic Transform (l-ECT). Unlike the raw Euler characteristic, l-ECT scans the local neighborhood in multiple directions, computing Euler characteristics of sublevel sets.
> >
> > Yes, I understand that.
> >
> > > We agree that this baseline [XGBoost on raw node features] is important and are happy to include the corresponding results in our revision. In the meantime, Table 6 already shows that l-ECT features are informative: as the number of directions used in the transform is reduced, classification performance often degrades substantially.
> >
> > Thank you for pointing me to this supplementary table. For 3 datasets out of 5 there is almost no difference there between the min and the max number of directions. So it remains possible that on these datasets XGBoost on raw features would perform as well. But I can appreciate that on 2 datasets there is indeed a pronounced difference.
> >
> > > While the experiment in Section 5.2 does serve as a sanity check, it also demonstrates that l-ECT provides a scalable alternative to alignment methods like Procrustes, which rely on costly SVD.
> >
> > Well... SVD is not that costly. For the tiny graphs in Figure 1 the cost is negligible. If you want to seriously claim that your procedure is faster than Procrustes rotation for very large graphs, then it should be directly demonstrated using runtime experiments. Otherwise it's fine with me if Section 5.2 gets reformulated as a sanity check.

---

> > > ### Author Response · Authors · 2025-04-04
> > >
> > > Dear Reviewer,
> > >
> > > Thank you for the clarification and the thoughtful follow-up.
> > > We fully understand your expectation that a rebuttal might also include new experimental results, and we appreciate you sharing this perspective. To complement our earlier conceptual, theoretical, and methodological responses, we are now able to share first results from ongoing experiments:
> > > * **GraphSAGE** consistently underperforms compared to our l-ECT-based method. For instance, on Computers we observe ~91.0%, and on Photo ~93.2%, both notably lower than our reported results.
> > > * **XGBoost trained solely on the raw features** performs considerably worse, e.g., ~86.6% on Computers (vs. ~92.2%) and ~92.1% on Photo (vs. ~94.9%).
> > > * **Additional preliminary node classification results** also confirm the strength of our method, exhibiting accuracies up to the following:
> > >
> > > | Setting                | Minesweeper | Tolokers   | Questions  |
> > > |------------------------|-------------|------------|------------|
> > > | l-ECT_1 + l-ECT_2      | 0.8699      | 0.8309     | 0.7639     |
> > > | l-ECT_1                | 0.8587      | 0.7936     | 0.7970     |
> > > | l-ECT_2                | 0.6233      | 0.8396     | 0.7572     |
> > >
> > > More experiments (those requested by Reviewers vCjY and DDk2) are currently running, and we are fully committed to incorporating all results into the camera-ready version.
> > >
> > > Finally, on Section 5.2: you're right—Procrustes is negligible for small graphs. Our aim was to highlight its impact in large or repeated alignment scenarios, but we’ll revise the section to better reflect this and make its role as a sanity check more explicit.
> > >
> > > We are grateful for your fair and constructive review. If the additional experiments and clarifications we’ve provided align with your expectations, we would kindly ask you to consider raising your score. Your support would make a meaningful difference at this stage.
> > >
> > > Best regards,
> > >
> > > The Authors

---

### Official Review · Reviewer_nYTM · 2025-03-10

**Overall Recommendation:** 4

**Summary:**

The paper introduces a local Euler Characteristic Transform (l-ECT), a local topology measure. l-ECT is an application of ECT for analysis of a neighborhood. Then, author apply it to enhance expressivity and interpretability of graph representations (mostly graph's nodes classification).
Authors identify crucial issues where GNN fail while l-ECT are useful. A rotation invariant l-ECT is provided. Theoretical guarantees are provided.

**Claims And Evidence:**

Main claims are:

> As our main contributions, we (i) construct l-ECTs in the context of embedded simplicial complexes (and graphs) and theoretically investigate their expressivity in the special case of featured graphs, (ii) empirically show that this expressivity positions l-ECTs as a powerful general tool for interpretable node classification, often superior to standard GNNs, and (iii) introduce an efficiently computable rotation-invariant metric based on l-ECTs that facilitates the spatial alignment of geometric graphs.

Claims are mostly convincing, except the "interpretability". The examples of interpretability are not provided.
The details on evaluation of rotation-invariant metric is not provided. How do you perform differentiation since the Euler characteristic is not differentiable?

**Essential References Not Discussed:**

--

**Experimental Designs Or Analyses:**

Experiments are sound. See questions above.

**Methods And Evaluation Criteria:**

Benchmarks are fine.
The performance of l-ECT on WebFB dataset is better that on common Planetoid. Do you have an explanation?

**Other Comments Or Suggestions:**

1. The definition of a simplical complex is quite clumsy mixing together geometrical and abstract simplical complexes.
2. Hyperlinks to proofs of theorems in Appendix is not provided in main text,
3. I assume that l-ECT is defined for a __geometric__ simplical complex, otherwise the definition (3) doesn't make sense.
Why you still use term "abstract simplicial complex" in line 113?
Consequently, l-ECT makes sense for graphs only if each vertex has attributes and a graph is mapped onto R^n.
If I am right, this shoud be stated more directly.

**Other Strengths And Weaknesses:**

**Strengths**

1. The paper introduced a novel concept, l-ECT and empirically proved its usefulness for node classification in graphs.

**Weaknesses**

1. The proposed method outperformed standard GCN, GAT, GIN but is still worse that modern methods dedicated to hetero-graphs (see Appendix).
2. If I understood correctly, l-ECT method is not applicable for graphs without features associated with nodes.

**Questions For Authors:**

See above.

**Relation To Broader Scientific Literature:**

Authors cite broader scientific literature, for example SOTA studies on topological deep learning and learning with hetero-graphs.

**Theoretical Claims:**

Yes, I have checked most of the proofs.

---

> ### Author Rebuttal · Authors · 2025-03-28
>
> Dear Reviewer,
>
> we sincerely thank you for the thoughtful and detailed feedback! **We are glad that you found our contributions novel and our empirical and theoretical results sound.** We especially appreciate your engagement with the proofs and your recognition of l-ECT’s usefulness for node classification. Below, we address your comments point by point.
>
> > The examples of interpretability are not provided.
>
> Thank you for raising this point. A discussion on the interpretability of l-ECT is included in Appendix A.2.1, where we show how feature importance of the underlying model naturally gives rise to interpretability. However, we agree that the main paper could more clearly showcase this aspect. **In the revised version, we will better illustrate how l-ECT enables interpretation of node roles and neighborhood structures.**
>
> > The details on evaluation of rotation-invariant metric is not provided.
>
> Section 5.2 evaluates our rotation-invariant metric for aligning spatial graph representations, showing it captures structural similarity under random rotations. **We’ll clarify this purpose in the text and add more details.**
>
> > How do you perform differentiation since the Euler characteristic is not differentiable?
>
> The Euler Characteristic Transform is made differentiable by embedding it into a continuous pipeline using soft (smoothed) filtrations. It doesn’t change the discrete nature of EC itself but creates a differentiable approximation around it. **We will clarify this in our revision.**
>
> > The performance of l-ECT on WebFB dataset is better that on common Planetoid. Do you have an explanation?
>
> The stronger performance of l-ECT on WebKB over Planetoid datasets reflects structural differences: WebKB graphs are heterophilic, where local structure matters more than feature similarity—making l-ECT well-suited—while Planetoid graphs are homophilic, favoring message-passing GNNs. **We’ll clarify this in the revision.**
>
> > The proposed method outperformed standard GCN, GAT, GIN but is still worse that modern methods dedicated to hetero-graphs (see Appendix).
>
> We agree that recent heterophily-specific methods are highly competitive on heterophilic benchmarks. However, our goal with l-ECT is not to design a specialized architecture optimized solely for heterophily, but rather to propose a general-purpose, interpretable representation that captures rich local structural information without message passing. What’s notable is that l-ECT achieves competitive performance despite being a non-aggregating and model-agnostic approach, and in many cases closes the gap with specialized models. We believe this highlights the complementary strengths of topological representations, particularly when combined with other methods. **We will emphasise this aspect further in our revision and substantiate it using additional experiments.**
>
>
> > If I understood correctly, l-ECT method is not applicable for graphs without features associated with nodes.
>
> You're right—l-ECT, as formulated, assumes node features or geometric embeddings. This is a common assumption in graph learning, where such features are typically available and important for downstream tasks like node classification. Our method therefore aligns well with standard benchmarks and real-world settings. **We'll clarify this assumption in the paper.**
>
> > The definition of a simplical complex is quite clumsy mixing together geometrical and abstract simplical complexes.
>
> Thank you for pointing this out—we completely agree. Our current presentation conflates geometric and abstract simplicial complexes. Since our method relies on geometric realizations, we’ll restrict the discussion to geometric simplicial complexes in the revised version. **We’ll clarify the terminology and focus solely on the relevant setting for l-ECT.** We appreciate your attention to this—it helps improve the clarity and precision of the exposition.
>
> > Hyperlinks to proofs of theorems in Appendix is not provided in main text
>
> Thanks for this suggestion. **In the revised version, we will include hyperlinks and explicit references from each theorem in the main text to its corresponding proof in the appendix.**
>
> > I assume that l-ECT is defined for a geometric simplical complex, otherwise the definition (3) doesn't make sense. Why you still use term "abstract simplicial complex" in line 113? Consequently, l-ECT makes sense for graphs only if each vertex has attributes and a graph is mapped onto R^n. If I am right, this shoud be stated more directly.
>
> This is a mistake, please see response above. **We will fix this in our revision.**
>
> We are grateful for your recognition of our contributions and your close reading of the manuscript. If the improvements we propose address your concerns, we would be very grateful if you would consider raising your score.
> Thank you once again for your constructive and thoughtful review, we’d be happy to address any other questions you might have.
>
> Best regards,
>
>  The Authors

---

> > ### Comment · Reviewer_nYTM · 2025-04-04
> >
> > Thank you for addressing my questions. I hope that the clarifications and an explanation how l-ECT is differentiated will be included in the camera-ready version.

---

> > > ### Author Response · Authors · 2025-04-04
> > >
> > > Thank you for your feedback. We're glad the clarifications were helpful and will ensure that the final version includes the necessary explanations and improvements.

---

### Official Review · Reviewer_DDk2 · 2025-03-13

**Overall Recommendation:** 3

**Summary:**

The paper introduces the Local Euler Characteristic Transform (l-ECT), a novel approach for graph representation learning that extends the Euler Characteristic Transform (ECT) to local neighborhoods. The key innovation is capturing local structural information in graphs without relying on conventional message-passing aggregation schemes used in Graph Neural Networks (GNNs). The authors theoretically show that l-ECTs can preserve local neighborhood information without loss and provide a rotation-invariant metric for spatial alignment. They empirically demonstrate that l-ECT-based approaches outperform standard GNNs on various node classification tasks, particularly on heterophilous graphs where traditional aggregation methods struggle.

## update after rebuttal
After checking rebuttal from different reviewers, I incline to keep my score.

**Claims And Evidence:**

Most claims are supported by theoretical analysis and empirical results.

**Essential References Not Discussed:**

LINKX (Lim et al., 2021) - A method that also bypasses message passing for heterophilous graphs
WRGAT (Suresh et al., 2021) - Addresses heterophily with rewiring techniques
ACM-GCN (Luan et al., 2022) - Reports state-of-the-art results on heterophilous benchmarks
Papers on graph topological operators like GraphGPS (Rampavsek et al., 2022)

**Experimental Designs Or Analyses:**

The experimental design is generally sound.

**Methods And Evaluation Criteria:**

The proposed methods are generally sound:
#### 1. Using l-ECTs to capture local neighborhood information is a novel approach that addresses limitations in traditional GNNs.
#### 2.The evaluation on both homophilous and heterophilous graph datasets is appropriate.
#### 3. The comparison with standard GNNs (GCN, GAT, GIN) and a heterophily-specific model (H2GCN) provides good context.

**Other Comments Or Suggestions:**

Actually, I think the experiments should pay more attention to the special properties the proposed method can demonstrate. I don't think heterophilous graph is a good target.

**Other Strengths And Weaknesses:**

### Strengths:

#### 1. Novel integration of topological data analysis techniques with graph learning
#### 2. Theoretically grounded approach with provable properties
#### 3. Strong performance on heterophilous graphs without specialized architecture

### Weaknesses:

#### 1. Limited exploration of computational efficiency - l-ECT computation could be expensive for large graphs
#### 2. Relationship between l-ECT parameters and performance is not thoroughly analyzed
#### 3. Limited comparison to recent heterophily-specific approaches

**Questions For Authors:**

1. How does the computational complexity of computing l-ECTs scale with graph size and neighborhood size? This seems critical for practical applications.

2. Could you clarify your notion of graph embedding using node features? How do you handle cases where the embedding might not preserve edge relationships?

3. What is the theoretical justification for l-ECTs working well on heterophilous graphs beyond the empirical results?

4. How do you address the scenario where nodes have identical feature vectors but different structural roles in the graph?

**Relation To Broader Scientific Literature:**

I am not sure about that.

**Theoretical Claims:**

Most claims are supported by theoretical analysis and empirical results.

---

> ### Author Rebuttal · Authors · 2025-03-28
>
> Dear Reviewer,
>
> we sincerely thank the reviewer for the careful and thoughtful evaluation of our work. We appreciate your recognition of our contributions, particularly the novelty of integrating topological data analysis into graph representation learning, and your acknowledgment of the soundness of our theoretical and empirical work. Below, we address your comments and questions in detail.
>
>
> > Limited exploration of computational efficiency—l-ECT computation could be expensive for large graphs.
>
> For a fixed node $x$, the computational complexity of its $\ell ECT_k(x)$ is:
>
> $$
> O\big(m \cdot l \cdot |N_k(x)|\big),
> $$
>
> where:
>
> - $m$ is the number of sampled directions,
> - $l$ is the number of filtration steps,
> - $|N_k(x)|$ is the number of vertices (or simplices) in the $k$-hop neighborhood of $x$.
>
> In other words, the complexity scales linearly with neighborhood size and sampling resolution. In practice, as shown in Appendix A.2.1, even a small subset of directions suffices for high accuracy, reducing runtime *considerably*. For large graphs, parallelization of l-ECT computation provides scaling of the procedure. **We will include a thorough discussion on computational complexity and improved implementations in our revision.**
>
> > Relationship between l-ECT parameters and performance is not thoroughly analyzed
>
> In Appendix A.2.1, we provide an ablation study on the number of sampled directions, showing how performance varies when using a smaller portion of directions. That said, we agree that further exploration with respect to other parameters such as the number of filtration steps l would be valuable. **We are happy to include these additional ablations in the revised version to provide a more complete picture of the trade-offs involved!**
>
> > LINKX (Lim et al., 2021) - A method that also bypasses message passing for heterophilous graphs WRGAT (Suresh et al., 2021) - Addresses heterophily with rewiring techniques ACM-GCN (Luan et al., 2022) - Reports state-of-the-art results on heterophilous benchmarks Papers on graph topological operators like GraphGPS (Rampavsek et al., 2022)
>
> We appreciate the references and agree these are valuable works that provide additional context. **We will discuss these references in our revised manuscript for an additional comparison!**
>
> > Actually, I think the experiments should pay more attention to the special properties the proposed method can demonstrate.
>
> This is an insightful observation. Our motivation for including heterophilous benchmarks was to highlight the method’s robustness in the absence of homophily, where traditional GNNs often degrade. We agree that l-ECT’s real strength lies in capturing local structure independent of homophily assumptions, yield broad utility. **In the revision, we will reframe the narrative to better emphasize the general-purpose nature of l-ECT.**
>
> > How does the computational complexity of computing l-ECTs scale with graph size and neighborhood size? This seems critical for practical applications.
>
> Please refer to the response above.
>
> > Could you clarify your notion of graph embedding using node features? How do you handle cases where the embedding might not preserve edge relationships?
>
> The embedding is created by assigning a node to the point in Euclidean space which is specified by its respective node feature vector. Edges are drawn between nodes if and only if there's an edge in the original graph. Since nodes may appear multiple times, edges might get lost by using this embeddings (see the paragraph before Thm.1). Therefore, in practice we use virtual edges to prevent such cases (i.e. an embedded node may occur multiple times as a neighbor). **We will clarify this subtlety in our revised paper, thanks for spotting that!**
>
>
> > What is the theoretical justification for l-ECTs working well on heterophilous graphs beyond the empirical results?
>
> The core limitation of message passing in heterophilous settings lies in its aggregation mechanism, which tends to blur dissimilar feature signals. Our method bypasses aggregation entirely, instead constructing a vectorized topological summary of a node’s neighborhood. From a theoretical standpoint, Thm. 1 shows that l-ECT_1 encodes sufficient information to reconstruct the feature vectors of a node’s neighbors without aggregation. This property is critical in heterophilous settings where the difference between features (not their average) is informative.
>
> > How do you address the scenario where nodes have identical feature vectors but different structural roles in the graph?
>
> Please see our response above.
>
> **We appreciate your detailed review and the encouragement regarding our theoretical and empirical contributions.** If our revisions adequately address your concerns, we would be very grateful if you would consider raising your overall score.
> Thank you again for your time and constructive feedback! Please let us know if you have any other questions or concerns.
>
> Best regards,
>
>  The Authors

---

### Official Review · Reviewer_vCjY · 2025-03-15

**Overall Recommendation:** 3

**Summary:**

The paper introduces a novel method called the Local Euler Characteristic Transform (ECT), designed to enhance graph representation learning by preserving critical local structures while maintaining global interpretability. ECT provides a lossless representation of local neighborhoods around graph nodes. This method, grounded in topological principles, uses Euler characteristic transformations to capture both structural and spatial information around each node, making it particularly useful for tasks like node classification. The authors argue that ECT can overcome limitations found in GNNs, especially in graphs with high heterophily, where aggregating neighboring information could obscure crucial differences. The paper also introduces a rotation-invariant metric based on ECT, which helps in spatially aligning data spaces, offering a practical advantage in comparing graph data.

**Claims And Evidence:**

Claims made in the submission are clear and convincing.

**Essential References Not Discussed:**

Please refer to my comment in Experimental Designs Or Analyses section.

**Experimental Designs Or Analyses:**

More heterophilic GNN baselines such as spectral-based models [2] [3] [4] and spatial-based models [5] [6] should be considered in the section 5. Besides, GraphSAGE is also a powerful baseline for heterophilic graphs.

[2] Bo et al. Beyond low-frequency information in graph convolutional networks. AAAI 2021.

[3] He et al. BernNet: Learning Arbitrary Graph Spectral Filters via Bernstein Approximation. NeurIPS 2021.

[4] Luan et al. Revisiting Heterophily For Graph Neural Networks. NeurIPS 2022.

[5] Li et al. Finding global homophily in graph neural networks when meeting heterophily. ICML 2022.

[6] Want et al. Powerful graph convolutional networks with adaptive propagation mechanism for homophily and heterophily. AAAI 2022.

**Methods And Evaluation Criteria:**

The Texas and Wisconsin datasets are relatively small, and previous studies [1] have indicated that the Chameleon and Squirrel datasets contain significant amounts of duplicate data, which undermines their reliability. So the above datasets are not suitable for evaluating heterophilic GNNs any more, and I suggest that authors should consider more datasets proposed by  [1] such as minesweeper, tolokers, questions.

[1] Platonov et al. A critical look at the evaluation of GNNs under heterophily: Are we really making progress? ICLR 2023.

**Other Comments Or Suggestions:**

NA.

**Other Strengths And Weaknesses:**

Strengths:

1. The paper is well-written, and the motivation behind the proposed approach is sound and clearly articulated.
2. The idea of extending the Euler Characteristic Transform to capture local graph structures is novel and intriguing.

Weaknesses:

1. Please refer to my comments in the sections above.
2. In Table 7 of Appendix A2.3, all the listed models should be accompanied by their corresponding references.

**Questions For Authors:**

NA.

**Relation To Broader Scientific Literature:**

This paper builds on limitations of losing local information by introducing the Local Euler Characteristic Transform (ECT), a method inspired by topological data analysis, which provides a topological fingerprint of local graph neighborhoods. The use of Euler characteristic transformations is rooted in previous works like Turner et al. (2014), who utilized Euler characteristics for global shape classification. However, unlike prior work, which focuses on global properties, this paper introduces a local variant, ECT, that preserves detailed neighborhood information while still maintaining global interpretability. This extension is crucial, especially for graphs with high heterophily, where local node characteristics may be more important than aggregated neighbor features.

**Theoretical Claims:**

I am not sure the correctness of the proposed proofs for the theorems.

---

> ### Author Rebuttal · Authors · 2025-03-28
>
> Dear Reviewer,
>
> we sincerely thank you for the thoughtful and constructive feedback.
> We are pleased that you found our paper to be well-written and our proposed method—the Local Euler Characteristic Transform (l-ECT)—to be novel and well-motivated. Below, we address your main concerns point-by-point and outline the changes we will make in the revised version.
>
> > The Texas and Wisconsin datasets are relatively small, and previous studies [1] have indicated that the Chameleon and Squirrel datasets contain significant amounts of duplicate data, which undermines their reliability. So the above datasets are not suitable for evaluating heterophilic GNNs any more, and I suggest that authors should consider more datasets proposed by [1] such as minesweeper, tolokers, questions.
> [1] Platonov et al. A critical look at the evaluation of GNNs under heterophily: Are we really making progress? ICLR 2023.
>
> Thank you for pointing this out. We agree with your assessment of Chameleon and Squirrel and appreciate the recommendation. While our focus was to provide a broad comparison with commonly used benchmarks in heterophilic GNN evaluation, we recognize the importance of newer, cleaner datasets. **In the revised version, we will include results on datasets like Minesweeper, Tolokers, and Questions as proposed in [1].**
>
> > More heterophilic GNN baselines such as spectral-based models [2] [3] [4] and spatial-based models [5] [6] should be considered in the section 5. Besides, GraphSAGE is also a powerful baseline for heterophilic graphs. [...]
>
> We appreciate the reviewer’s recommendation and agree that these are strong and relevant baselines in the context of heterophily-specific architectures. However, we wish to emphasize that our approach is not only proposed as a specialized architecture for heterophilic graphs, but rather as a general-purpose representation method that provides a robust and interpretable alternative to neighborhood aggregation. Our focus is on illustrating the effectiveness of l-ECT as a complementary or standalone representation, rather than competing directly with task-specific model architectures. That said, we agree that GraphSAGE is a widely adopted and illustrative baseline for generalization across homophilic and heterophilic settings. **We will include GraphSAGE in our revised experiments, and clarify the scope of our contribution with respect to architecture design for heterophilic graphs.**
>
> > In Table 7 of Appendix A2.3, all the listed models should be accompanied by their corresponding references.
>
> Thank you for spotting this oversight. **We will revise Table 7 to include full citations for each model.**
>
> We thank the reviewer again for acknowledging the **novelty, motivation, and clarity of our work.** We believe that the l-ECT offers a fundamentally new perspective on graph representation learning—particularly by preserving geometrical-topological information at the local scale, a signal that is often diluted by aggregation-based models.
> If our revisions adequately address your concerns, we would be grateful if you would consider raising your overall score. We are happy to address any additional questions.
>
> Best regards,
>
> The Authors

---

### Decision · Program_Chairs · 2025-05-01

**Decision:**

Accept (poster)

**Comment:**

This paper introduces the local Euler Characteristic Transform (Diss-l-ECT), a novel, and theoretically grounded approach to graph representation learning that enhances node classification performance, especially on heterophilic graphs. The method is well-motivated, clearly presented, and shows strong empirical results across diverse benchmarks, outperforming standard GNNs in several settings. While the paper would benefit from deeper analysis and intuition about why Diss-l-ECT is effective and a direct comparison with XGBoost on raw features, the core contribution is technically sound and compelling. I recommend acceptance.